PERSPECTIVE

# Using Cartesian Doubt To Build a Sequencing-Based View of Microbiology

Braden T. Tierney,[a,b,c,d] Erika Szymanski,[e] James R. Henriksen,[f] Aleksandar D. Kostic,[b,c,d] Chirag J. Patel[a]

aDepartment of Biomedical Informatics, Harvard Medical School, Boston, Massachusetts, USA
bSection on Pathophysiology and Molecular Pharmacology, Joslin Diabetes Center, Boston, Massachusetts, USA
cSection on Islet Cell and Regenerative Biology, Joslin Diabetes Center, Boston, Massachusetts, USA
dDepartment of Microbiology, Harvard Medical School, Boston, Massachusetts, USA
eDepartment of English, Colorado State University, Fort Collins, Colorado, USA
fMr. Fusion Inc., Fort Collins, Colorado, USA

**ABSTRACT** The technological leap of DNA sequencing generated a tension between modern metagenomics and historical microbiology. We are forcibly harmonizing the output of a modern tool with centuries of experimental knowledge derived from culture-based microbiology. As a thought experiment, we borrow the notion of Cartesian doubt from philosopher Rene Descartes, who used doubt to build a philosophical framework from his incorrigible statement that "I think therefore I am." We aim to cast away preconceived notions and conceptualize microorganisms through the lens of metagenomic sequencing alone. Specifically, we propose funding and building analysis and engineering methods that neither search for nor rely on the assumption of independent genomes bound by lipid barriers containing discrete functional roles and taxonomies. We propose that a view of microbial communities based in sequencing will engender novel insights into metagenomic structure and may capture functional biology not reflected within the current paradigm.

**KEYWORDS** Cartesian doubt, microbial genetics, microbial species concept, microbiome

Cartesian doubt—beginning with radical skepticism and moving forward with as few external assumptions as possible—can be used to reconceive our approaches to microbiome science, potentially avoiding biases and conflicts stemming from centuries of culture-based microbiology. In 1641, Rene Descartes published his *Meditations on First Philosophy*, in which he upended and tossed aside past philosophical thought by asking "How do we know what is true?" (1). We propose similarly rethinking microbial communities as revealed via DNA sequencing, reimagining what microbial life may be instead of assuming what it is based on existing understandings of taxonomy, microbial genomes, or other culture-centric paradigms.

Consider metagenomic sequencing as an incarnation of Anton van Leeuwenhoek's microscope: peering through its "lens," what do we "see?" A FASTA file certainly does not display the discrete particles Anton van Leeuwenhoek described: sequencing is a lens foreign to historical microbiologists' view of microbes. Nevertheless, microbiome science routinely maps sequencing reads to "species" and "core" or "accessory" genes. Why restrict metagenomes to this paradigm, overlaying modern tools with centuries of single-species-centric experimental work rooted in physical observation? (2, 3). In light of the potential for epistemic conflicts between culture-based and sequencing-based knowledge, can the field establish an analytic frame that integrates these distinct perspectives?

Gaps between metagenomics (4) and historical microbiology illustrate why microbiome scientists should reconsider our core assumptions (Fig. 1A)—though the field

Address correspondence to Braden T. Tierney, btierney@g.harvard.edu, Aleksandar D. Kostic, Aleksandar.Kostic@joslin.harvard.edu, or Chirag J. Patel, chirag_patel@hms.harvard.edu.

should not adopt an ahistorical view. Rather, researchers should acknowledge that contemporary analyses can be biased by prior experiments. For example, phylogenies (5) employed as buckets for sequence data amalgamate physiological (e.g., via Bergey's manual and numerical taxonomy) and genetic markers (e.g., 16S sequence similarity) built through specific, now-historical perceptions of microbial life (6–10). This can constrain our understanding of microbiome biology (Fig. 1B). Additionally, "complete" genomes are defined through gold standard cutoffs that prioritize genes on the basis of their presence in previously assembled sequence data (11). Ecosystem-spanning signals extraneous to our current frameworks, like horizontal gene transfer (HGT) or evolutionary drift, look like noise to a framework built for monocultures and not communities (Fig. 1C). Assembly-based methods for genome discovery may therefore artificially bias gene content in organisms—or functions—with high rates of HGT. Further, the functional roles of similar sequences are often defined through global percent identity cutoffs, despite sequence not necessarily correlating to function (12). Finally, bio- and geochemical reactions exist in multiple spatial and temporal structures that may not be membrane bound within discrete cells. Overall, microbiologists constructed paradigms to cohere with pure cultures; a sequencing-centered approach to metagenomics unconstrained by pure-culture-based paradigms provides an exciting opportunity to rethink assumptions about the organization of microbial life.

While it is impossible to truly disregard a preconceived framework derived from hundreds of years of experimentation, Cartesian doubt can address epistemological conflicts between observations (i.e., raw data) from microbial communities and the paradigms (i.e., theory) used to interpret them. Consider working from the following axiom: a metagenome is captured in a data structure representing complete "sequencing of microbial DNA"—base pair order (e.g., reads), chemical structure (e.g., methylation), and spatial structure (e.g., via Hi-C [high-throughput chromosome conformation capture]). In other words, we hypothesize that DNA sequencing will advance such that it operates at any read length with increased resolution for sequence chemistry and structure. An unprejudiced view of this idealized sequencing data would allow the field to, at least temporarily, abnegate the paradigms that currently bind us and identify novel metagenomic structure.

Microbiome pattern identification is initially an algorithmic task. Modern approaches to metagenomic data analysis today discard ostensibly junk reads that, for example, do not map to draft genomes or assemble cleanly; unbiased approaches should first aim to minimize discarded data to avoid biological signal loss (13–16). Methodologically, numerous tools are used to "project" complex, unordered data into human-readable, low-dimensional space (Fig. 2A). These tools stem mostly from computer science and natural language processing, and some have already been applied to metagenomic data (17, 18). One simple method may be to collect k-mers in individual sequence reads across time, collapsing them into highly correlated clusters. Researchers could also consider using extensions of vector-based sequence projection methods, colored de Bruijn graphs, or metabolic network strategies (19–24). These approaches will advance analyses unconstrained by the paradigm of individual cells containing individual genomes. However, methods and data structures (e.g., databases indexing the k-mers of the Sequence Read Archive) should be selected carefully, as different questions mandate different approaches.

Would a sequence-first approach revise our view of genomes, genes, or codons? Any algorithm effectively parsing read data will identify conservation in sequence. Consider an approach that identifies consistent patterns in DNA base pair order. This may identify codons, as they are conserved and nonrandom. Comparison across reads could uncover alternate coding schemes as variations within this pattern (25). Perhaps longer reads would recover genes with little sequence divergence. Genes with high divergence would likely not cocluster; however, conserved motifs may. Biologists might have to further reconsider the fundamental units of microbial genetics (26, 27)

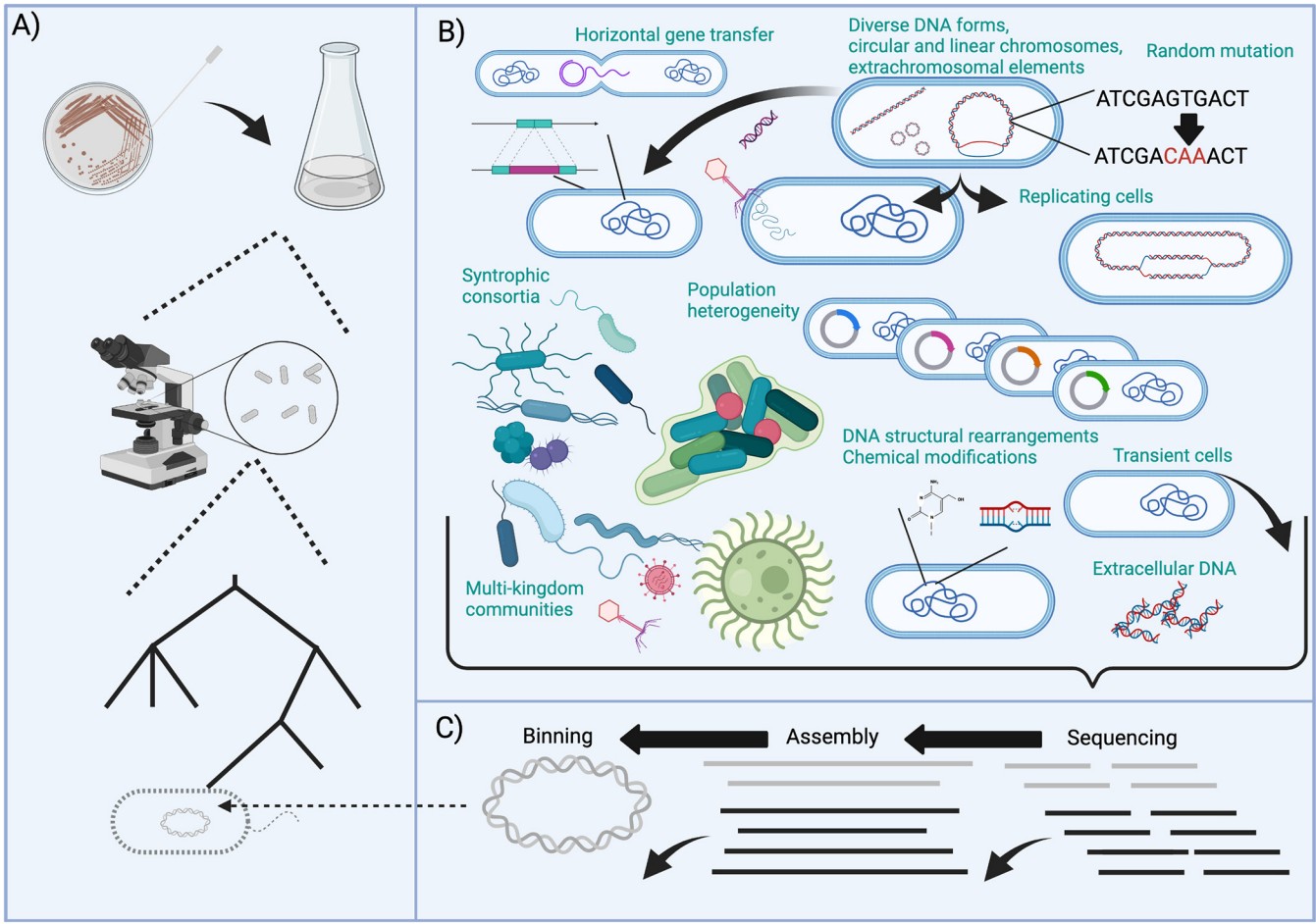

**FIG 1** The existing paradigm of microbiome science. (A) Our historical view of microbes originates from what is culturable. Bacteria, specifically, have been mostly observed in clonal isolation and are assumed to have measurable cell-based genomes that can be hierarchically grouped by phylogenetics. (B) Microbiome scientists (generally) use DNA sequencing to investigate a complex, multikingdom microbial community that is changing across space and time through a series of complex interactions that are not well represented by this framework, including horizontal gene transfer, cell replication, and spontaneous mutation. (C) To build a sense of microbial (bacterial in this case) genomes, researchers, for example, assemble sequencing reads into contigs and bin contigs into "complete," phylogenetically annotated, genomes. The figure was generated with BioRender.com.

For example, would core and accessory genes—or other patterns entirely—exist at higher levels of genomic organization (e.g., across metagenomes instead of genomes)?

Analyses based upon existing paradigms may also be limited in their capacities to capture genomic temporal variation. No microbial genome (or genome within an organism in any kingdom of life) is static across time and space. Replication forks, structural rearrangements, CRISPR spacer acquisition and loss, HGT, and plasmids will yield continually "incomplete" genomes, even if a single read could capture a contiguous unit of DNA. Unbiased pattern identification that aims to discover fundamental, spatiotemporally consistent (or inconsistent) metagenomic units will align our view of microbial genomics along an entirely new axis, redefining our perspective on microbiomic temporal modulation (Fig. 2B).

We hypothesize that unbiased approaches to sequence analysis would yield a continuum of sequence-based conservation: sequence substructures (e.g., motifs, genes conserved at high percent identity) that represent emergent biology, not necessarily tied to pathways, genes, or genomes. These substructures could, however, be periodically cooccurring, dynamic (or temporally periodic) elements that may, for example, be environmentally dependent or affect ecosystem-level functions. This "periodic table of metagenomic elements," which would minimize assumptions about meaningful versus noise reads, could provide increased insight into latent metagenomic structure.

Historically, important biology has been overlooked (e.g., the kingdom of archaea,

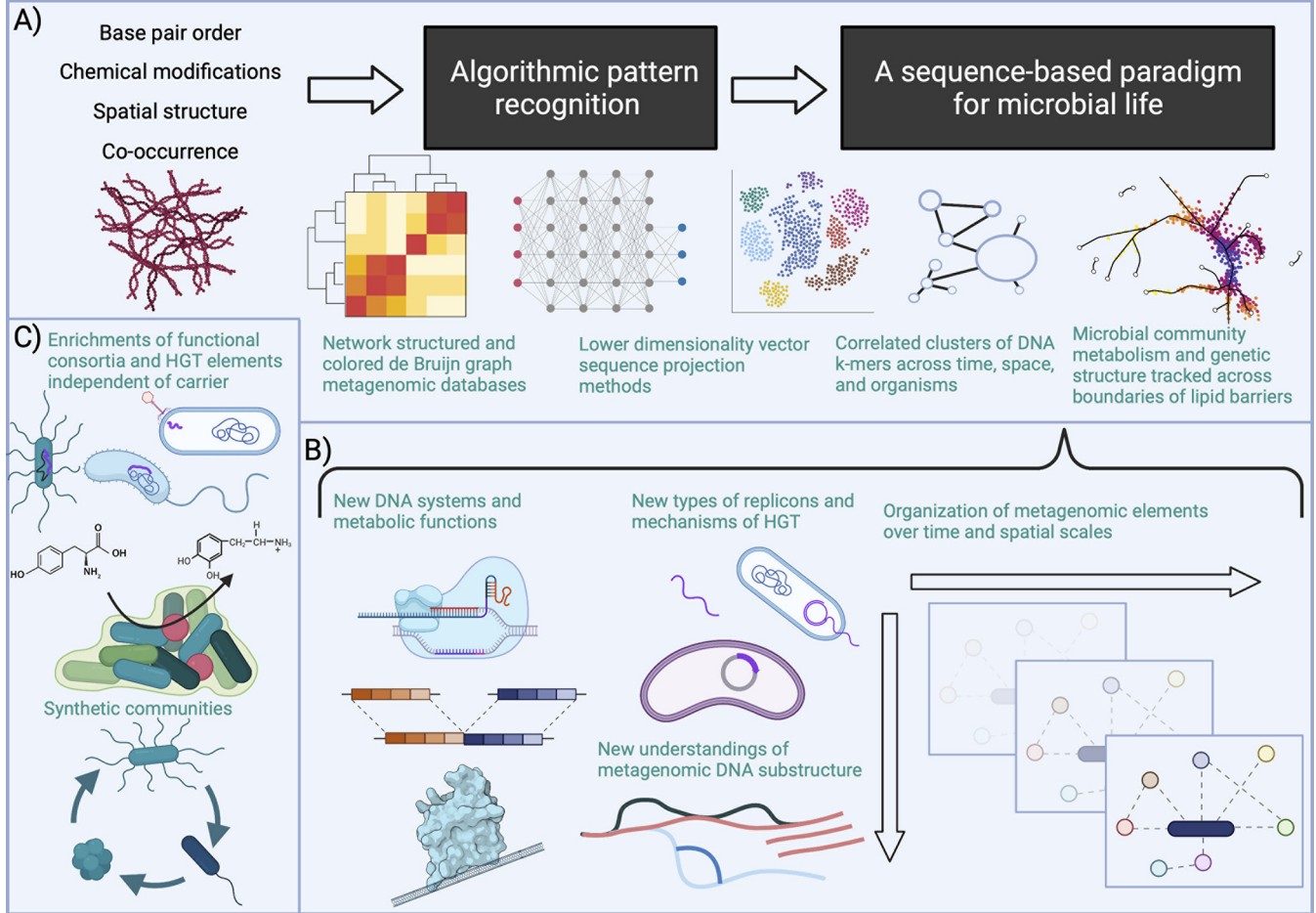

**FIG 2** Discovering new frameworks with Cartesian doubt. We propose using Cartesian doubt to consider sequencing data (referring to a range of multi-omic technologies) and how unbiased pattern recognition (A) can result in a cell-agnostic, sequencing-based paradigm that would be complementary to but unguided by the experimental history of microbiology (B). Combined with novel wet-lab techniques working within this new view of metagenomics, microbiome scientists could thereby reveal potentially unknown biology outside the scope of our current framework (C). The figure was generated with BioRender.com.

noncoding DNA, noncanonical amino acids) because technology was not designed to detect it or because assumptions limited the capacity to interpret biological signals, even at times construing such signal as contamination (28, 29). The reads that float between disparate genomes (or nodes that cannot be resolved in *de novo* assemblies) should be treated as signals, not hidden by forcing resolution or filtered out by data handling. Hundreds of thousands of reference microbial genomes derive from metagenome-assembled genomes (MAGs) built using culture-based "gold standards," which may exclude genes (e.g., conjugation systems) (30) or may amplify genomic features like random mutation, HGT, or doubling under the guise of new genome discovery. Finally, considering beyond base pair order-focused approaches will facilitate the incorporation of alternative sequencing data (e.g., Hi-C) into our understanding of metagenomic communities, their meta-phenotypes (e.g., colonization resistance), and their diversity (e.g., bacteria, fungi, and viruses).

Applying Cartesian doubt to microbiome science has numerous applications to rethinking the rules of life for microbiomes, ranging from our view of metagenomic DNA substructure to the microbial species concept to the tools used to work with metagenomes. We challenge the scientific and funding communities to pursue three efforts in particular. First, since microbial metabolism is not bounded by lipid barriers in a community setting, neither should our metagenomic paradigm. Scientists need to extend (20, 22, 23, 31, 32) and create new algorithms that integrate across sequencing

modalities and consider metagenomes as greater than sums of their parts. Second, theory and empirical data collection (i.e., experimental practice) need to inform each other. Currently, the field's assumptions constrain methodological development. If microbiological theory were less historically biased, further (33–35) wet-lab techniques for operating on different units of microbial life could be developed, perhaps extending on current synthetic community work but relying more on enrichments of functional consortia or independent HGT elements (Fig. 2C). Data interpretation methods are also needed, such as theoretical modeling (20) and algorithms operating on k-mers and microbiome metabolism. Finally, "gold standards" must be defined only in the context of a particular research question, avoiding claims regarding universality, as doing so obscures assumptions that may be invalid in context (e.g., >95% sequence identity when comparing genes or "complete" genomes).

Minimizing assumptions will add complementary insight to current paradigms while adding richness to our understanding of the functional organization of complex microbiomes. Indeed, Cartesian doubt's true power is accommodating many different perspectives, not necessarily unveiling some grand truth, but rather adjusting reference points through complementary scientific lenses. The historical model of microbiology has gotten us extremely far, and its value cannot be overlooked. However, while we all stand on the shoulders of giants, it is occasionally prudent to consider the ground beneath our feet.

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
