## [Reviewer comments · mSystems]

Using Cartesian doubt to build a sequencing-based view of microbiology

Braden Tierney, Erika Szymanski, James Henriksen, Aleksandar Kostic, and Chirag Patel

Corresponding Author(s): Chirag Patel, Harvard Medical School

Review Timeline:

Submission Date:	May 7, 2021
Editorial Decision:	July 5, 2021
Revision Received:	August 3, 2021
Editorial Decision:	September 3, 2021
Revision Received:	September 7, 2021
Accepted:	September 23, 2021

Editor: Linda Kinkel

Reviewer(s): The reviewers have opted to remain anonymous.

Transaction Report:

DOI: <https://doi.org/10.1128/mSystems.00574-21>

July 5, 2021

Dr. Chirag J Patel
Harvard Medical School
Biomedical Informatics
10 Shattuck St
Boston, MA 02115

Re: mSystems00574-21 (Using Cartesian doubt to build a sequencing-based view of microbiology)

Dear Dr. Chirag J Patel:

Thank you for submitting your manuscript to mSystems. We have completed our review and I am pleased to inform you that, in principle, we expect to accept it for publication in mSystems. However, acceptance will not be final until you have adequately addressed the reviewer comments.

Overall, this is a really fine set of ideas to bring to our collection! Both reviewers have thoughtful suggestions for improving the piece (and I echo the suggestion to include a citation for Jo Handelsman's work), and I encourage you to incorporate these into your revisions. In addition, I provide many comments below. Please do not be put off by the many comments—we are working closely with all our author teams to try to put together pieces that are accessible to our very broad intended audience, and to capture big ideas as effectively as possible. I am happy to discuss any of these suggestions further, via email or Zoom, and respond to any questions that you may have. I have indicated 'accept with minor modifications', trusting that you and your team will make your best effort to address my recommendations and those of the reviewers. We are excited to have this piece as part of our special issue!

I recommend that you rewrite the Abstract after completing your revisions. The 'tension' does not exist because of the leap forward. The tension exists bc DNA sequencing of communities and individuals provides data of a fundamentally different NATURE than traditional pure culture studies.

Make the 'preconceived notions' and 'experimental knowledge' more explicit earlier in this paragraph. You are explicitly arguing that biological organization of microbes/microbial communities can be (should be, may be better) considered at a different level of organization: the complex, messy, amazing, complete metagenome. This is what the aggregate DNA sequence data represents. Consider here what is lost by constraining our analyses to MAG, and what can be gained by focusing on metagenomic structure itself that may more accurately represent functional biology of microbiomes. ALSO, moving beyond stamp collecting. Driving a new organizational approach is likely to lead to novel insights, and to continue to move us beyond bins and boxes to new ways of thinking about the 'integrated microbiome'. This is itself a NEW paradigm, which is core to your FINAL SENTENCE OF THE ABSTRACT. Strengthen this or make it more prominent (earlier) in the abstract and again in the introduction!

Introduction: Avoid too much colloquial presentation: Omit "One thing is certain", "say, Louis Pasteur's view of microbes", omit "it might as well represent different forms of life."

The use of 'we' is perhaps sub-optimal. Consider generalizing more to our community, E.g. l. 34: "Nevertheless, most microbiome scientists focus first on mapping sequencing read to 'species', ..."

l. 35: "Why do we restrict microbial life to these particular data structures, overlaying a modern tool with centuries of experimental work reliant on physical observation of microorganisms?" MAKE THIS SENTENCE MUCH MORE CREATIVE. Ultimately, this IS about our BIOLOGICAL paradigm, not simply an analytical paradigm, but you don't take the BIOLOGICAL paradigm on head-first. DO IT. Is the conflict between culture-based and sequencing-based KNOWLEDGE vs. APPROACHES? This is an essential issue that you need to resolve! Isn't the challenge how we integrate these....which is perhaps as much about individual-based vs. community-based perspectives for analysis/integration of complex data forms? AND being very explicit in what we are MISSING with a primarily MAG-focused analytical approach. Make a passionate argument for the biological context from which your argument flows.

l. 41: Is it tensions or is it gaps?

Figure 1 is important, but recommend radically revising the legend to make it more informative (also omit 'circular!'). Note that not all microbiologists are proponents of a Linnean taxonomic tree—there are many alternatives that should be noted or at least cited in passing. Do modern metagenomics analyses aim to enforce (or reinforce?) this paradigm---or rather are modern metagenomics analyses constrained by this paradigm (I think this is one of your major messages!!!---folks aren't doing this because they want their analyses to stink, but there is a failure of both imagination and analytical frameworks and

capacities....so, like a river, folks stick to the channel. MISTAKE!). For `B', you provide a very detailed description, which may be more appropriately integrated into the main body of the text? (though also useful here, this is your call). I don't now that all folks seek to `place static genomes into a phylogenetic system'---this may be a little harsh. Also, the current legend for "C" seems to be one of the most succinct statements of your goal in this manuscript---in which case you should put this in more places---"unsupervised algorithmic approaches", is that strictly what you mean? Do you prefer multiple approaches, ranging from unsupervised to diverse strategies for supervising, with explicit clarification of both the assumptions and constraints of each them? We don't know what we don't know, and we won't find it if we are ONLY constrained to analyses that are structured fundamentally by what we already `know' or think that we know. For microbiomes this is a HUGE issue!! Note that it is likely still an unanswered question whether or not unsupervised algorithmic approaches will `more accurately represent the (measured-is this word necessary?) biology therein'. I think this is a proposition/hypothesis that you are asserting-but could couch by noting that at the very least may offer a completely different, potentially more or more realistic or complementary or... representation of the biology of microbiomes?

I. 43-45, change phrasing "and genetic markers defined by centuries-old (note it is barely centuries!) perceptions (understanding?) of the biology of single organisms, generally in pure culture." Is this actually about taxonomy, or our fundamental understanding of microbial biology? Perhaps be more explicit in pulling out different challenges here: i) taxonomic-centered/organized view of microbes/microbial genomes/microbiomes constrains understanding! (we have long known this is a terrible idea, microbes exist in coordinated communities with extensive signaling and chemical interactions that mediate the activities of the members; ii) lines 45-47 capture a distinct problem: we only consider what we already know from existing data---which precludes inclusion of unknown genes; iii) horizontal gene transfer and evolutionary drift are especially vulnerable to exclusion due to (ii); iv) this is especially problematic for organisms OR FUNCTIONS that are highly susceptible to genetic drift, horizontal gene transfer, or strong or rapid selection (which may be most interesting and important to know!); v) function-sequence relationships are assumed with a significant lack of corresponding biological information; vi) functions/reactions need not be membrane bound (clarify how this IS or IS NOT distinct from (i)). FINALLY, strengthen the final sentence here: rather than having constructed pure cultures over time to fit or match our paradigms, isn't it in fact that we have constructed our paradigms to fit a (shockingly small and feeble collection of data based upon) an `individual-organism'-based approach. DESPITE all of these major flaws to the individual-organism-based approach...biological, analytical,

I. 59: Cartesian Doubt is a good framing device, what about including Aristotle's metaphysics, "the whole is different from the sum of the parts" (which seems certainly to require a distinct paradigm) incorporated into lines 59-65. One of your big points is that the parts are not equal to the whole, and/or that the ways that parts are defined fundamentally limits what we can know, and/or that a focus on parts alone misses the bigger story/the biology of the system. Our former paradigms were constrained by the experimental approaches that were essentially limited to and focused on measuring individual organisms/taxa as part of a bigger whole. With our new sequencing approaches, we have the opportunity to broaden our paradigm to ask the bigger /better questions/do better biology. Note also that the sequence-based approaches are way more effective in capturing data from non-cultured/non-culturable taxa which have been largely beyond the existing paradigms bc we haven't been able to access them effectively as single cultures.

I. 63: I'm not sure that `radical skepticism', if you are advocating that knowledge is impossible, is a productive lens for us? I like to think that scientists believe that some level of knowledge IS possible, and that rethinking and reimagining as new ways of knowing or measuring become available is critical to getting closer to what may be conceived of as truth.

I. 67: Bias seems to me to be too weak a word here---you are suggesting a complete revision of the framework for DNA-based microbiome analyses and thinking. Also this paragraph could use a closing sentence that ties this all up---this is about walking away from limiting paradigms AND anticipating the higher-order (methylation, spatial position) power that sequencing data is providing and will continue to provide at higher quality.

I. 79: define junk? Why in quotation marks? What is the value of pre-partitioning---and can you simplify this sentence to include both examples in one parenthetical---(e.g. long vs. short reads, free vs. lipid-bound DNA)? NOTE Hi-C is presented both as HiC and Hi-C in the text---be consistent.

I. 87: should this be a new, separate paragraph, or does it continue the `Unbiased pattern identification...' paragraph?

I. 91: It would be helpful if you could suggest some tools and/or corresponding questions. Can you finish this paragraph with some suggestions of the data type/structure that would be of great biological interest in considering microbiomes in a non-organismal or non-binned, unbiased pattern identification approach? OR if you are waiting to do this until later (e.g.

I. 112), I suggest you make a stronger transition in I. 95: "How might a sequence-first approach alter our current view of genomes, genes, or codons?"

I. 100-101: why specifically might this force a reconsideration of what we imagine as a `gene'? Should you be using the term `gene' in the previous sentence? Is this really just sequences with high divergence are unlikely to be grouped (if the unbiased pattern identification is `working', of course they should not be grouped!).

I. 102-103: This sentence is tossed out with little evaluation---but is this good or bad? Is this something that is potentially lost with

this metagenomic approach? Or do concepts of 'core' or 'accessory' become re-established at a higher/larger organizational scale---at the scale of the community/metagenome rather than the organism or the binned/organismal genome?

I. 105-110: As noted by reviewers, remove references to circular genomes, as not inclusive of all bacteria and omits fungi (also valued members of the microbiome). Also, I am not sure that the idea of 'complete' genomes is important in this paragraph? If we are seeking to move beyond the individual genome as a focal point, does this matter? Are individual genomes more static across time and space than metagenomes/microbiomes? How important is the variability of the genome/metagenome in time and space to our understanding of the microbiology/microbiomes, and does the unbiased pattern identification approach offer some unique or superior solution to developing this understanding? It seems that there is either something deeper here, or that this doesn't add much to the piece.

I. 112: what is a 'grounded' approach---can you define 'grounded'? It seems that an unbiased pattern identification approach is probably less grounded than using MAG to analyze metagenomic data?

I. 115-117: Can you make this sentence BIGGER (and it is not clear that you need to raise core and accessory genes here): These foundational metagenomic elements, and their dynamics in relation to more variable genetic elements among microbiomes and environments could provide critical insights into the biology of microbiome functioning (make this all about using metagenomic data to unlock unlock the biology of microbiomes).

I. 199: Omit "This unbiased new 'taxonomy'"---don't need to loop back to taxonomy as a concept here, but the periodic table of metagenomic elements is an exciting idea. I'm not sure what you mean by 'the temporal continuum of metagenomic life', but metagenomic structures and their variation in time and space is a powerful idea.

I. 123: because our technology wasn't designed to detect it or because our assumptions limited our capacity to interpret important biological signals. For example, the reads that

I. 126: The 'exploding MAG' problem seems a little bit of a red herring, the key issue is that gold standard cutoffs may amplify genomic noise. Can you highlight this point, and make the exploding MAG issue a secondary or de-emphasized point? Moreover, aren't you implicitly arguing that these things were ASSUMED to be 'noise', but that they are NOT explicitly noise-HGT, for example, could be important information about the structure of the metagenome?

I. 130: Smooth the transition: Moreover, moving away from approaches that that emphasize only base-pair order will facilitate the incorporation of alternative sequencing technologies and data types into our understanding of microbiomes. (note: if distinguishing between functionally discrete yet genetically identical cells....is this something that goes back to categorizing binned, whole genomes/MAGs? This seems like a step backwards in the key point(s) that you are trying to make in the paper).

I. 134: a great sentence in which to use 'microbiome science' rather than 'microbial science'. Can you give MORE than just the single application of metagenomic structure/substructure insights?

I. 138: Are you really arguing for a rethink of TAXONOMY? Or are you suggesting a radical revision of what taxonomy means (which is what I think you are arguing for), in which case I encourage you to articulate this more clearly. Do you mean a taxonomy of the community as an aggregate unit? What ideas can you touch on in the literature to illustrate this more clearly?

I. 140: There is a LOT of research in synthetic communities that I think builds to what you are talking about here, including both wet lab/experimental work and whole genome-based metabolic modeling (which can/should be reenvisioned from a metagenomic place). Are you suggesting something specific beyond the extensive work in this area at present? Clarity would help.

I. 143: Rather than a avoiding 'universal gold standards', perhaps what is more critical is defining the standards AND defining the underlying assumptions and justification for the standards in relation to core questions. Standards can facilitate progress, but can also impede progress. The sweet spot is figuring out what they are accomplishing one versus the other---which generally depends upon the question. Notable, your line 146-147 falls back into the box of genomic analyses, but as far as I understand the paper a central point is that microbiomes can and should be explored using non-genome-constrained analyses for our field to both break out of a constraining paradigm and make big advances in fundamental understanding.

I. 149: what is meant by the idea of 'application' here? It doesn't seem to be a practical application that you are referring to, but rather the ideas/questions that are being addressed? And notable, you are not advocating a paradigm-free approach, but really you are suggesting an explicit alternative paradigm to provide a distinct and important new perspective on the organization and structure of microbiomes. Certainly this new paradigm will both contribute wonderful new insights, but will also suffer from its own assumptions, but the complementarity of these paradigms is ultimately (in my view) the strength of our science. It seem to me that you are calling out the prospect of knowledge based upon: i) a single paradigm; and/or ii) the lack of consideration of the assumptions implicit or explicit in the paradigm that is being used to organize knowledge; and/or iii) a lack of recognition of the explicit limitation of a single paradigm for capturing all understanding/knowledge of a microbiome; and iv) a lack of recognition that new forms of data can empower new paradigms that may displace, or may complement, or may augment, or may empower

existing paradigms. It seems to me that this isn't an either/or but really strong keeping our eyes open to multiple perspectives. We can accommodate, integrate, contrast, and query multiple data types using distinct paradigms to form a more perfect understanding (knowledge), e pluribus unum for science? Out of many (or at least multiple) paradigms, can we find the best knowledge? I would encourage you to take a BIGGER VIEW to your closing paragraph to inspire the community to adopt a bigger vision for microbiome science.

Use MICROBIOMES instead of micro-organisms or microbiology in places where possible throughout (recognizing that sometimes microbiology or microorganisms makes more sense).

Also, what about fungi? Are the arguments you make here equally valid for fungi within microbiomes? Can you state this clearly somewhere in the manuscript? We would prefer that all of our pieces be as inclusive as possible.

Preparing Revision Guidelines

For complete guidelines on revision requirements for your article type, please see the journal Article Types requirement at <https://journals.asm.org/journal/mSystems/article-types>. **Submissions of a paper that does not conform to mSystems guidelines will delay acceptance of your manuscript.**

Sincerely,

Linda Kinkel

Editor, mSystems

Journals Department
Reviewer comments:

Reviewer #1 (Comments for the Author):

This article provides a useful reminder for microbiome researchers to think outside of the "individual-organism" box, that is, to transition from thinking of a microbiome as the additive sum of individuals to thinking of it as a synergistic collective. The authors focus specifically on nucleic sequence data, as these widely available data are more prone to linkages to the originating cells than non-sequence-based data such as metabolomes. This article is harkening back to the original concept of metagenomes put forth by Jo Handelsman, which was to think of the community itself as a meta-organism with the metagenome representing the

collective. This is a powerful concept and one that bears repeating, reminding and presenting from new viewpoints. This article is useful in that it highlights some of the hurdles that are distracting us from this goal. For example, the authors highlight the current emphasis on assembly, which is highly problematic in complex communities; these efforts are directing time and resources away from efforts to extract information from the collective and often result in incomplete and likely highly error-prone assemblies. The authors also highlight how the desire for standards, which are a mechanism for increasing data credibility and quality, can also enforce assumptions that limit the potential for new discoveries as to the way that DNA is moved, shared and changes within a microbiome.

Overall, the article is a useful thought-piece encouraging scientists to question all assumptions regarding how microbes live and function within communities and to incorporate this questioning in their approaches to analyzing metagenome data. Such critical thinking is fundamental to good science, but given the potential for new discoveries in microbiome science and our tendency to fall back on what we already know of microbes, the authors' advice to use Cartesian doubt is a useful reminder to researchers. This is particularly needed for microbiomes since most of our current understanding of microbes comes from observing pure cultures (not communities) in liquid culture (not biofilms) under ideal conditions for growth (not resource limited) and in a constant environment (not with environmental shifts characteristic of most natural habitats). Hence, most of what we know may simply not be applicable in the microbiomes that we are studying.

A few suggestions:

1. The authors assume that microbial genomes are circular. This assumption has been long dispelled, since some microbes have linear chromosomes (e.g., *Borrelia*, *Streptomyces*, *Rhodococcus*) and even both circular and linear chromosomes (e.g., *Agrobacterium*). I would recommend deleting the references to circular genomes on lines 64, 105 and 157, and modifying "circularized" to "complete" on line 108.
2. I agree that minimizing discarded data is critical (I find that this is true even with comparative genomics of closely-related strains, so it would definitely be true for a community). However, it is difficult to see how "pre-partitioning [reads] by length" would help minimize discarded data (line 81). This should be clarified.
3. I was surprised that the authors did not discuss the approach of comparing metagenomic reads, in the absence of assembly, to databases. Lines 95-101 discuss finding patterns in the sequences, including conserved motifs, but deriving meaningful information from these patterns requires knowledge. A read-based rather than assembly-based approach is being taken by some researchers and somewhat avoids this pigeonholing of all reads into "genes". Are there hurdles to this approach that should be addressed in their recommendations on lines 134-147? For example, are there changes/expansions to our databases that would improve the quality or speed of doing read-based comparisons?
4. Lastly, I did not understand how wet lab approaches for microbial communities, which I presume means amplifying and preserving these communities for experimentally repeatable studies, could be an "extension" of precision editing technologies. Rather, I can envision that the precision editing technologies themselves could be an outcome of success with the wet lab approaches.

And a few minor editorial suggestions:

5. Line 24. Delete "or" - "...that neither search for [or] nor rely on..."
6. Line 41. The sentence needs a verb. Tensions between metagenomics and historical microbiology evidence [illustrate? Indicate?] why we should reconsider our core assumptions.
7. Line 65. Suggested revision "...not assuming the veracity of taxonomy, of genomes, or frankly of what we think we know about microbial life"
8. Line 81. Suggested revision "it may be worth pre-partitioning by length or another feature such as biological source" or "it may be worth pre-partitioning by features such as biological source".

Reviewer #2 (Comments for the Author):

The manuscript by Tierney and colleagues is an interesting Perspective article that encourages microbiologists to think differently about the field - to approach it from a DNA-centric approach rather than an organismal approach. The manuscript is couched in a philosophical framework calling researchers to use Cartesian doubt/radical doubt/radical skepticism. Overall, although I appreciate that this is a shorter format of manuscript, I felt the arguments were too abbreviated to be clear and did not adequately represent the current state of the field.

The title and manuscript seem to use "Cartesian doubt", "radical doubt", and "radical skepticism" interchangeably without a good

solid definition the first time the phrases are used. I gather that "Cartesian doubt" is the most widely used term in philosophy. I would encourage the authors to use that phrase throughout and to give a simple definition the first time it is used. I needed to do a google search to understand the title.

The authors assume a very specialized audience throughout the manuscript that is highly attuned to techniques and issues that are commonly used with genomic and metagenomic sequencing. I consider myself someone that knows much of what the authors are talking about and still found much of the manuscript was way too much "insider baseball" (e.g. de bruijn graphs, Hi-C, exploding MAG problem, etc). The manuscript would really benefit from the authors expanding their examples and being more clear about what they are arguing. The manuscript needs a thorough edit with the eyes of someone that is not so immersed in the field.

I was a bit lost throughout this manuscript as to what the authors were really saying. So many of the examples of problems in this Perspective are because of methods - missing the Archaea, junk DNA, etc. To the authors' first example of traditional microbiology, even Van Leeuwenhoek's predecessors missed the Bacteria because of the limits of technology. I would even argue that a microscope is not that far removed from software that processes a FASTA file. Van Leeuwenhoek had to grind lenses to magnify the samples using a very sensitive process that was not reproducible to other scientists at the time - many thought he was fabricating results! (doi: 10.1098/rstb.2014.0344) The parallels are similar with bioinformatics software for processing DNA sequences. To me, the problems are with picking the best methods and questioning the assumptions of those methods rather than with the underlying biological hypothesis we are applying.

Isn't the problem the authors are outlining with our use of methods? I'm not convinced that Science is not sufficiently self correcting. I think it sees through these problems. Unfortunately, the authors see Science as "half-empty" in that regard. As the authors rightly acknowledge there are plenty of examples of where our algorithms have been founded on incorrect assumptions. Yet the paragraph detailing this (L77-85) has few citations and the citations that are included describe methods. They might consider doi: 10.1126/science.1142490 as an example for where "contaminating bacterial DNA" was actually a Wolbachia genome embedded in an insect genome. Although they mention that we didn't know about Archaea until relatively late, we did find out about them through both cellular and molecular approaches. Even later we realized that we were missing phyla of Archaea through microscopy and molecular approaches (DOI: 10.1073/pnas.0914470107) - but Science self corrected itself in all of these examples. Again, I'm left wondering what the authors were really arguing for in this manuscript.

I am generally confused by their claimed results of a "thought experiment". We know that linear bacterial (and viral) genomes exist (doi: 10.1111/j.1462-2920.2007.01328.x). I'm not sure that we could figure out codons from only DNA sequences without amino acid sequences. At a minimum, a citation should be provided here. Geneticists are already reconsidering what a gene is, without regard for metagenomics or even bacteria (doi: 10.1101/gr.6339607)

I think the authors are also somewhat selective in their rejection of the cellular model of science. They discuss "tensions" (L41) but only indicate problems with the traditional approach. If there is a tension then there must be problems on both sides. The manuscript is very on-sided in its critique. Are there no benefits from the traditional approach? The paragraph starting at L119 seems to argue for treating communities as bags of genes, disregarding the importance of cellular envelopes. Although their argument for lack of importance of cellular structure is based on metabolism being widespread across a community, cellular structures could still be important. I also think of cellular structures being important for other characteristics like pathogenesis and toxin production. Do the authors see no benefit to considering cells as the atomic microbial level?

The final paragraph starts with, "Allowing our view of microbial communities to be guided by application and not paradigm-bias..." Yet, I would argue that Science moves forwards because we have paradigms - hypotheses - that are continually refined. If everything is operationally or practically defined, Science cannot move forward or self correct.

To Dr. Kinkel, the Reviewers, and the mSystems Editorial Team,

We thank you for your constructive feedback on our manuscript. We are grateful for the time you took to review it and have endeavoured to integrate your comments thoroughly. As a result of your contributions, we believe the work is now much improved, and we are excited to share it with you. Broadly speaking, we made three major overhauls:

- 1) We now clearly state the approach and the theoretical framing for the argument (i.e. definition of Cartesian doubt) in the first paragraph. We additionally have walked back overly technical, “insider baseball” language wherever possible to ensure that our manuscript will be approachable by a large audience.
- 2) We have included a number of specific examples regarding approaches and algorithms, the possible outcomes from our analysis, and we have dropped extraneous arguments that overall detracted from our manuscript (e.g. excessive discussion of Metagenome-Assembled-Genomes and core/accessory genes).
- 3) Finally, we have increased the precision of our language throughout regarding the outcomes of our thought experiments and, most importantly, our recommendations to the field moving forward.

We additionally have totally overhauled our figure and figure legend to the point where we have 2 different figures now that communicate all the main points of our text.

We provide point by point response to reviewer comments below in blue, citing passages from the text where relevant. When necessary, we annotate attention to key phrases by highlighting in yellow. We provide a tracked-changes manuscript and also reproduce the entirety of Reviewer comments (while adding in a numbering scheme to make them easily referenceable).

Sincerely, and on behalf of the authors,

Chirag J Patel and Aleksandar D Kostic

Editorial comments:

1.1 I recommend that you rewrite the Abstract after completing your revisions. The 'tension' does not exist because of the leap forward. The tension exists bc DNA sequencing of communities and individuals provides data of a fundamentally different NATURE than traditional pure culture studies.

We have updated the abstract accordingly, reproducing it here:

“The technological leap of DNA sequencing generated a tension between modern metagenomics and historical microbiology. We are forcibly harmonizing the output of a modern tool with centuries of experimental knowledge derived from culture-based microbiology. As a thought experiment, we borrow the notion of Cartesian doubt from philosopher Rene Descartes, who used doubt to build a philosophical framework from his incorrigible statement that “I think therefore I am.” We aim to cast away preconceived notions and conceptualize microorganisms through the lens of metagenomic sequencing alone. Specifically, we propose funding and building analysis and engineering methods that neither search for nor rely on the assumption of independent genomes bound by lipid barriers containing discrete functional roles and taxonomies. We propose that a view of microbial communities based in sequencing will engender a sense of metagenomic structure that more closely represents functional biology and not our current notions of it.”

1.2 Make the 'preconceived notions' and 'experimental knowledge' more explicit earlier in this paragraph. You are explicitly arguing that biological organization of microbes/microbial communities can be (should be, may be better) considered at a different level of organization: the complex, messy, amazing, complete metagenome. This is what the aggregate DNA sequence data represents. Consider here what is lost by constraining our analyses to MAG, and what can be gained by focusing on metagenomic structure itself that may more accurately represent functional biology of microbiomes. ALSO, moving beyond stamp collecting. Driving a new organizational approach is likely to lead to novel insights, and to continue to move us beyond bins and boxes to new ways of thinking about the 'integrated microbiome'. This is itself a NEW paradigm, which is core to your FINAL SENTENCE OF THE ABSTRACT. Strengthen this or make it more prominent (earlier) in the abstract and again in the introduction!

In addition to strengthening the abstract and the (now 3rd) paragraph, where we list extensively many of the tensions between culture and sequencing based microbiome science, we additionally have moved up our definition of Cartesian doubt and its (broad) application to open the piece, clarifying the goal from the outset. We believe doing so further addresses this point. Specifically, this first paragraph now reads:

“Cartesian doubt -- beginning with radical skepticism and moving forward with as few external assumptions as possible -- can be used to reconstruct our current practice of microbiome science, addressing biases and conflicts stemming from centuries of culture-based microbiology. In 1641, Rene Descartes published his “Meditations on First Philosophy,” in which he upended and tossed aside past philosophical thought by asking: “How do we know what is true?”(1) We propose similarly rethinking microbial communities via DNA sequencing,

reimagining what microbial life may be instead of assuming what it is on the basis of existing taxonomy, microbial genomes, or other culture-centric paradigms.

”

1.3 Introduction: Avoid too much colloquial presentation: Omit "One thing is certain", "say, Louis Pasteur's view of microbes", omit "it might as well represent different forms of life."

We have cleaned the text here, it now reads:

“A FASTA file certainly does not display the discrete particles Anton Van Leeuwenhoek described: sequencing is a lens foreign to historical microbiologists’ view of microbes.”

1.4 The use of `we' is perhaps sub-optimal. Consider generalizing more to our community, E.g. I. 34: "Nevertheless, most microbiome scientists focus first on mapping sequencing read to `species', ..."

We perused the manuscript and generalized uses of the word “we,” except where we specifically refer to the author team (e.g. we claim, we hypothesize).

1.5 I. 35: "Why do we restrict microbial life to these particular data structures, overlaying a modern tool with centuries of experimental work reliant on physical observation of microorganisms?" MAKE THIS SENTENCE MUCH MORE CREATIVE. Ultimately, this IS about our BIOLOGICAL paradigm, not simply an analytical paradigm, but you don't take the BIOLOGICAL paradigm on head-first. DO IT. Is the conflict between culture-based and sequencing-based KNOWLEDGE vs. APPROACHES? This is an essential issue that you need to resolve! Isn't the challenge how we integrate these....which is perhaps as much about individual-based vs. community-based perspectives for analysis/integration of complex data forms? AND being very explicit in what we are MISSING with a primarily MAG-focused analytical approach. Make a passionate argument for the biological context from which your argument flows.

We now attempt to make these sentences more about the fundamental biology and integrating the historical microbiology and metagenomics:

“Why restrict metagenomes to this paradigm, overlaying modern tools with centuries of single-species-centric experimental work reliant on physical observation?(2, 3) In light of the potential for epistemic conflicts between culture-based and sequencing-based knowledge, can the field establish an analytic frame that integrates these multiple perspectives?”

1.6 I. 41: Is it tensions or is it gaps?

We agree “gaps” makes sense in this context and have modified accordingly.

1.7 Figure 1 is important, but recommend radically revising the legend to make it more informative (also omit 'circular!'). Note that not all microbiologists are proponents of a Linnean taxonomic tree--there are many alternatives that should be noted or at least cited in passing. Do modern metagenomics analyses aim to enforce (or reinforce?) this paradigm---or rather are modern metagenomics analyses constrained by this paradigm (I think this is one of your major messages!!!---folks aren't doing this because they want their analyses to stink, but there is a failure of both imagination and analytical frameworks and capacities....so, like a river, folks stick to the channel. MISTAKE!).

We have radically revised our figure to accommodate these points and also fully communicate the purpose of the entire manuscript and details therein. We have now split the figure into 2 (reproduced below), with one focused on the current paradigm and one focused on how we can apply Cartesian doubt, replete with technical examples.

We have removed the reference to Linnaean taxonomy specifically as well as the circular genome point. We cite a reference to a perspective on the broad array of taxonomic methods used today in the text (<https://www.sciencedirect.com/science/article/pii/S0966842X20303279>).

For 'B', you provide a very detailed description, which may be more appropriately integrated into the main body of the text? (though also useful here, this is your call). I don't now that all folks seek to 'place static genomes into a phylogenetic system'---this may be a little harsh.

We have walked back the "static genome" statement here (see below). We choose to leave this in the legend now due to the word limit and the ability to communicate more in the revised figures, though we believe the updated manuscript captures this information much better than the initial draft.

Also, the current legend for "C" seems to be one of the most succinct statements of your goal in this manuscript---in which case you should put this in more places---"unsupervised algorithmic approaches", is that strictly what you mean? Do you prefer multiple approaches, ranging from unsupervised to diverse strategies for supervising, with explicit clarification of both the assumptions and constraints of each them? We don't know what we don't know, and we won't find it if we are ONLY constrained to analyses that are structured fundamentally by what we already 'know' or think that we know. For microbiomes this is a HUGE issue!! Note that it is likely still an unanswered question whether or not unsupervised algorithmic approaches will 'more accurately represent the (measured-is this word necessary?) biology therein'. I think this is a proposition/hypothesis that you are asserting-but could couch by noting that at the very least may offer a completely different, potentially more or more realistic or complementary or... representation of the biology of microbiomes?

We updated legend C (now Figure 2) to make it more specific and clear regarding what our thought experiment's output would be. We also walked back the "accurately" point, and slightly modified the text in the figure regarding the "completely different/complementary" point.

We reproduce both the updated figures and legend here:

Figure 1: The existing paradigm of microbiome science. A) Our historical view of microbes originates from what is culturable. Bacteria, specifically, have been mostly observed in clonal isolation and are assumed to have measurable cell-based genomes that can be hierarchically grouped by phylogenetics. B) Microbiome scientists (generally) use DNA sequencing to investigate a complex, multi-kingdom microbial community that is changing across space and time through a series of complex interactions that are not well represented by this framework, including horizontal gene transfer, cell replication, and spontaneous mutation. C) To build a sense of microbial (bacterial in this case) genomes, researchers, for example, assemble sequencing reads into contigs and bin contigs into “complete,” phylogenetically-annotated, genomes.

Figure 2: Discovering new frameworks with Cartesian doubt. We propose using Cartesian doubt to consider sequencing data (referring to a range of multi-omic technologies) and how A) unbiased pattern recognition can B) result in a cell-agnostic, sequencing-based paradigm that would be complementary to but unguided by the experimental history of microbiology. Combined with C) novel wet-lab techniques working within this new view of metagenomics, microbiome scientists could thereby reveal potentially unknown biology outside the scope of our current framework.

1.8 I. 43-45, change phrasing "and genetic markers defined by centuries-old (note it is barely centuries!) perceptions (understanding?) of the biology of single organisms, generally in pure culture." Is this actually about taxonomy, or our fundamental understanding of microbial biology? Perhaps be more explicit in pulling out different challenges here: i) taxonomic-centered/organized view of microbes/microbial genomes/microbiomes constrains understanding!

We agree that this sentence -- as we as much of our initial draft -- put too much weight on the phylogeny and the word "taxonomy" -- as a result, we have now crafted our argument to clearly supersede phylogenies alone and instead consider our fundamental understanding of microbial communities and genetic substructure. We cite one example of doing this here, however the shift of focus away from phylogenies specifically permeates the revised manuscript.

“Rather, researchers should acknowledge that contemporary analyses are biased by prior experiments. For example, phylogenies(5) employed as buckets for sequence data amalgamate physiological (e.g. via Bergey’s manual and numerical taxonomy) and genetic markers (e.g. 16S sequence similarity) built through specific, now-historical perceptions of microbial life.(6–10) These constrain our understanding of microbiology (Figure 1B).”

(we have long known this is a terrible idea, microbes exist in coordinated communities with extensive signaling and chemical interactions that mediate the activities of the members; ii) lines 45-47 capture a distinct problem: we only consider what we already know from existing data---which precludes inclusion of unknown genes; iii) horizontal gene transfer and evolutionary drift are especially vulnerable to exclusion due to (ii); iv) this is especially problematic for organisms OR FUNCTIONS that are highly susceptible to genetic drift, horizontal gene transfer, or strong or rapid selection (which may be most interesting and important to know!); v) function-sequence relationships are assumed with a significant lack of corresponding biological information; vi) functions/reactions need not be membrane bound (clarify how this IS or IS NOT distinct from (i)).

To these points above, we also now make the point regarding functions, writing:

“Metagenome-spanning signals extraneous to our current paradigm, like horizontal gene transfer (HGT) or evolutionary drift, look like noise to a framework built for monocultures and not communities (Figure 1C). Assembly-based methods for genome discovery may therefore artificially bias gene content in organisms -- or functions -- with high rates of HGT.”

We also took the “insider baseball” out of this line, removing a reference to de bruijn graphs.

FINALLY, strengthen the final sentence here: rather than having constructed pure cultures over time to fit or match our paradigms, isn't it in fact that we have constructed our paradigms to fit a (shockingly small and feeble collection of data based upon) an `individual-organism'-based approach. DESPITE all of these major flaws to the individual-organism-based approach...biological, analytical,

“Overall, microbiologists have constructed paradigms to cohere with pure cultures; however, they haven’t constructed microbiomic paradigms in the same way, yielding an exciting opportunity to rethink assumptions about microbial life’s organization.”

1.9 I. 59: Cartesian Doubt is a good framing device, what about including Aristotle's metaphysics, "the whole is different from the sum of the parts" (which seems certainly to require a distinct paradigm) incorporated into lines 59-65. One of your big points is that the parts are not equal to the whole, and/or that the ways that parts are defined fundamentally limits what we can know, and/or that a focus on parts alone misses the bigger story/the biology of the system. Our former paradigms were constrained by the experimental approaches that were essentially limited to and focused on measuring individual organisms/taxa as part of a bigger whole. With our new sequencing approaches, we have the opportunity to broaden our paradigm to ask the

bigger /better questions/do better biology. Note also that the sequence-based approaches are way more effective in capturing data from non-cultured/non-culturable taxa which have been largely beyond the existing paradigms bc we haven't been able to access them effectively as single cultures.

We agree that you capture here one of the fundamental components of our piece re: the whole being greater than the sum of the parts. We've now emphasized in the second to last paragraph of the paper re: one of the three forward looking recommendations we make:

“First, since microbial metabolism is not bounded by lipid barriers in a community setting, neither should our metagenomic paradigm. Scientists need to extend(20, 22, 23, 31, 32) and create new algorithms that integrate across sequencing modalities and consider metagenomes as greater than sums of their parts.”

1.10 I. 63: I'm not sure that 'radical skepticism', if you are advocating that knowledge is impossible, is a productive lens for us? I like to think that scientists believe that some level of knowledge IS possible, and that rethinking and reimagining as new ways of knowing or measuring become available is critical to getting closer to what may be conceived of as truth.

Given this comment as well as the comments of Reviewer 3 (specifically 3.1, 3.2), it is clear that our use of the phrase radical skepticism as interchangeable with Cartesian doubt was confusing. As a result, we now provide a clear definition of Cartesian doubt up front and no longer attempt to use any synonyms throughout the text.

To the broader point we absolutely agree that science is capable of generating valid knowledge and in no way want to make the contrary argument. Rather, we're aiming here to take the epistemological stance, considering how we view the relationship between knowledge and data. To make this clearer to the reader, we state it up front:

“While it is impossible to truly eliminate a preconceived framework derived from hundreds of years of experimentation, Cartesian doubt can address epistemological conflicts between observations (i.e. raw data) from complex microbial communities versus the paradigms (i.e. theory) they are forced into.”

1.11 I. 67: Bias seems to me to be too weak a word here---you are suggesting a complete revision of the framework for DNA-based microbiome analyses and thinking. Also this paragraph could use a closing sentence that ties this all up---this is about walking away from limiting paradigms AND anticipating the higher-order (methylation, spatial position) power that sequencing data is providing and will continue to provide at higher quality.

To strengthen the point, we now open this paragraph with:

“While it is impossible to truly disregard a preconceived framework derived from hundreds of years of experimentation”

And we close this paragraph with:

“An unprejudiced view of this idealized sequencing data would allow the field to, at least temporarily, abnegate the paradigms that currently bind us and identify novel metagenomic structure.”

1.12 I. 79: define junk? Why in quotation marks? What is the value of pre-partitioning---and can you simplify this sentence to include both examples in one parenthetical---(e.g. long vs. short reads, free vs. lipid-bound DNA)? NOTE Hi-C is presented both as HiC and Hi-C in the text---be consistent.

We now have removed the quotation marks and clarify that junk refers to reads that are thrown out for one reason or another:

“Modern approaches to metagenomic data analysis today discard ostensibly junk reads that, for example, do not map to draft genomes or assemble cleanly; unbiased approaches should first aim to minimize discarded data to avoid biological signal loss.”

We agree also that the pre-partitioning idea was unclear at best and have removed it. We also fixed the notation regarding Hi-C.

1.13 I. 87: should this be a new, separate paragraph, or does it continue the ‘Unbiased pattern identification...’ paragraph?

We agree upon further inspection that it should be merged with the prior paragraph and have done so.

1.14 I. 91: It would be helpful if you could suggest some tools and/or corresponding questions. Can you finish this paragraph with some suggestions of the data type/structure that would be of great biological interest in considering microbiomes in a non-organismal or non-binned, unbiased pattern identification approach? OR if you are waiting to do this until later (e.g.

I. 112), I suggest you make a stronger transition in I. 95: "How might a sequence-first approach alter our current view of genomes, genes, or codons?"

We now mention a series of tools and provide citations for them, as well as recommending a simple data structure:

“Methodologically, numerous tools are used to “project” complex, unordered data into human-readable, low-dimensional space (Figure 2A). These stem mostly from computer science and natural language processing, and some have already been applied to metagenomic data.(17, 18) One simple approach may be to collect k-mers in individual sequence reads across time, collapsing them into highly correlated clusters. Researchers could also consider

using extensions of vector-based sequence projection methods, colored de Bruijn graphs, or metabolic network strategies(19–24). The resulting data would allow analyses unconstrained by the paradigm of individual cells containing individual genomes. However, methods and data structures (e.g. a database indexing the k-mers of the Sequence-Read-Archive) should be selected carefully, as different questions mandate different tools.”

Here are the references for the tools we mention here (some of which we cite additionally in the second to last paragraph):

Genome graphs and the evolution of genome inference

Paten B, Novak AM, Eizenga JM, Garrison E

Genome Res., 2017

DNA sequence representation without degeneracy

Yau SS, Wang J, Niknejad A, Lu C, Jin N, Ho YK

Nucleic Acids Res., 2003

Predicting Nash equilibria for microbial metabolic interactions

Cai J, Tan T, Joshua Chan SH

Bioinformatics, 2020

Success of Alignment-Free Oligonucleotide (k-mer) Analysis Confirms Relative Importance of Genomes not Genes in Speciation and Phylogeny

Forsdyke DR

arXiv [q-bio.PE], 2019

Viral Sequence Identification in Metagenomes using Natural Language Processing Techniques

Abdelkareem AO, Khalil MI, Elbehery AH, Abbas HM

bioRxiv, 2020

High-quality genome sequences of uncultured microbes by assembly of read clouds

Bishara A, Moss EL, Kolmogorov M, Parada AE, Weng Z, Sidow A, Dekas AE et al.

Nat. Biotechnol., 2018

Continuous Embeddings of DNA Sequencing Reads and Application to Metagenomics

Menegaux R, Vert JP

J. Comput. Biol., 2019

Practical application of self-organizing maps to interrelate biodiversity and functional data in NGS-based metagenomics

Weber M, Teeling H, Huang S, Waldmann J, Kassabgy M, Fuchs BM et al.

ISME J., 2011

Metabolic network-guided binning of metagenomic sequence fragments

Biggs MB, Papin JA
Bioinformatics, 2016

DNA sequence representation without degeneracy
Yau SS, Wang J, Niknejad A, Lu C, Jin N, Ho YK
Nucleic Acids Res., 2003

Genome graphs and the evolution of genome inference
Paten B, Novak AM, Eizenga JM, Garrison E
Genome Res., 2017

Exploring neighborhoods in large metagenome assembly graphs reveals hidden sequence diversity
Titus Brown C, Moritz D, O'Brien MP, Reidl F, Reiter T, Sullivan BD
bioRxiv, 2019

To the specific relevant data structure, while we do believe it varies depending on the question, we also now describe in the next paragraphs a more specific “type” of metagenomic element that might be interesting to consider, those that are conserved across different metagenomes (as opposed to genomes) or those that occur across time. We write:

“Biologists might well have to further reconsider the fundamental units of microbial genetics.(26, 27) For example, would core and accessory genes -- or other patterns entirely -- exist at higher levels of genomic organization (e.g. across metagenomes instead of genomes)?”

“Unbiased pattern identification that aims to discover fundamental, spatiotemporally consistent (or inconsistent) metagenomic units would align our paradigm of microbial genomics along an entirely new axis, redefining our perspective on microbiomic temporal modulation (Figure 2B).”

1.15 I. 100-101: why specifically might this force a reconsideration of what we imagine as a `gene'? Should you be using the term `gene' in the previous sentence? Is this really just sequences with high divergence are unlikely to be grouped (if the unbiased pattern identification is `working', of course they should not be grouped!).

We agree the use of the word “gene” here was confusing -- we have adjusted accordingly:

“Biologists might well have to further reconsider the fundamental units of microbial genetics.”

We additionally cite this manuscript as an example of such reconsideration: Evelyn Fox Keller -- Keller, E. F. Genes, Genomes, and Genomics. *Biol Theory* **6**, 132–140 (2011).

1.16 I. 102-103: This sentence is tossed out with little evaluation---but is this good or bad? Is this something that is potentially lost with this metagenomic approach? Or do concepts of `core'

or 'accessory' become re-established at a higher/larger organizational scale---at the scale of the community/metagenome rather than the organism or the binned/organismal genome?

We agree that this sentence was tossed out and not clear -- we now specifically ask the following question to clarify what we mean:

"For example, would core and accessory genes -- or other patterns entirely -- exist at higher levels of genomic organization (e.g. across metagenomes instead of genomes)?"

1.17 I. 105-110: As noted by reviewers, remove references to circular genomes, as not inclusive of all bacteria and omits fungi (also valued members of the microbiome).

The repeated emphasis on circular genomes was an oversight, and we've now removed all references to this concept. We also now explicitly mention non-bacterial microbes elsewhere in the manuscript.

We write:

"Finally, considering beyond base-pair-order-focused approaches will facilitate alternative sequencing (e.g. Hi-C) into our understanding of metagenomic communities, their meta-phenotypes (e.g. colonization resistance), and their diversity (e.g. bacteria, fungi, and viruses)."

"No microbial genome (of any kingdom of life)..."

Also, I am not sure that the idea of 'complete' genomes is important in this paragraph? If we are seeking to move beyond the individual genome as a focal point, does this matter? Are individual genomes more static across time and space than metagenomes/microbiomes? How important is the variability of the genome/metagenome in time and space to our understanding of the microbiology/microbiomes, and does the unbiased pattern identification approach offer some unique or superior solution to developing this understanding? It seems that there is either something deeper here, or that this doesn't add much to the piece.

We also strongly agree with the sentiment of this line of questioning and have removed the emphasis on complete genomes. To indicate the broader point we are trying to make, we cite a new preprint out of Jill Banfield's group where they identified a unique kind of genomic element that goes beyond our traditional concept of genomes and thereby could be identified by an unbiased approach. The citation occurs after this sentence:

"Historically, massive swaths of biology have been missed (e.g. the kingdom of archaea, non-coding DNA, non-canonical amino acids) because technology was not designed to detect it or because assumptions limited scientific capacity to interpret important biological signals, even at times construing biological signal as contamination.(28, 29)"

We agree “completeness” is not the important part here. We now emphasize that unbiased pattern identification in the context spatiotemporal fluctuation in metagenomes would shift how we currently view microbial genomics:

“Unbiased pattern identification that aims to discover fundamental, spatiotemporally consistent (or inconsistent) metagenomic units would align our paradigm of microbial genomics along an entirely new axis, redefining our perspective on microbiomic temporal modulation.”

1.18 I. 112: what is a ‘grounded’ approach---can you define ‘grounded’? It seems that an unbiased pattern identification approach is probably less grounded than using MAG to analyze metagenomic data?

Grounded is perhaps another example of “insider baseball.” It is an epistemological term that perhaps is too niche (Glaser & Strauss. *Discovery of Substantive Theory: A Basic Strategy Underlying Qualitative Research. The American Behavioral Scientist* **8**, (1965)). As a result, we have replaced its use in the manuscript with more easily understood, if less-specific, vernacular (e.g. “unbiased”).

1.19 I. 115-117: Can you make this sentence BIGGER (and it is not clear that you need to raise core and accessory genes here): These foundational metagenomic elements, and their dynamics in relation to more variable genetic elements among microbiomes and environments could provide critical insights into the biology of microbiome functioning (make this all about using metagenomic data to unlock unlock the biology of microbiomes).

We have removed the reference to core/accessory genes and aimed to expand the depth of this sentence:

“We hypothesize that an unbiased approach to analyzing sequence data would yield a continuum of sequence-based conservation: sequence substructures (e.g. motifs, genes conserved at high percent identity) that represent emergent biology, not necessarily tied to pathways, genes, or genomes. These could, however, be periodically co-occurring, dynamic (or temporally periodic) elements that may, for example, be environmentally dependent or affect ecosystem-level functions.”

1.20 I. 199: Omit "This unbiased new ‘taxonomy’-don’t need to loop back to taxonomy as a concept here, but the periodic table of metagenomic elements is an exciting idea. I’m not sure what you mean by ‘the temporal continuum of metagenomic life’, but metagenomic structures and their variation in time and space is a powerful idea.

We agree and have omitted the reference to taxonomy as requested (as we have throughout the manuscript to keep readers focused on the bigger picture outside of just, say, the species concept).

The sentence now reads:

“This “periodic table of metagenomic elements,” which would minimize assumptions about meaningful versus noise reads, could provide increased insight into latent metagenomic structure.”

1.21 I. 123: because our technology wasn't designed to detect it or because our assumptions limited our capacity to interpret important biological signals. For example, the reads that

We agree this point about assumptions is important and have added it in.

1.22 I. 126: The “exploding MAG” problem seems a little bit of a red herring, the key issue is that gold standard cutoffs may amplify genomic noise. Can you highlight this point, and make the exploding MAG issue a secondary or de-emphasized point? Moreover, aren't you implicitly arguing that these things were ASSUMED to be “noise”, but that they are NOT explicitly noise-HGT, for example, could be important information about the structure of the metagenome?

We now clean up our discussion of MAGs, reducing the overly-detailed technological focus and the overstated “exploding MAG” problem while addressing how we talk about “genomic noise”. We write:

“Hundreds of thousands of reference microbial genomes derive from Metagenome-Assembled-Genomes (MAGs) built using culture-based “gold standards,” which may exclude genes (e.g. conjugation systems)(30), or may amplify genomic features like random mutation, HGT, or doubling under the guise of new genome discovery.”

1.23 I. 130: Smooth the transition: Moreover, moving away from approaches that that emphasize only base-pair order will facilitate the incorporation of alternative sequencing technologies and data types into our understanding of microbiomes. (note: if distinguishing between functionally discrete yet genetically identical cells....is this something that goes back to categorizing binned, whole genomes/MAGs? This seems like a step backwards in the key point(s) that you are trying to make in the paper).

We agree that this point was out of place and have integrated a lightly modified version of your recommend change into the manuscript:

“Finally, considering beyond base-pair-order-focused approaches will facilitate alternative sequencing (e.g. Hi-C) into our understanding of metagenomic communities, their meta-phenotypes (e.g. colonization resistance), and their diversity (e.g. bacteria, fungi, and viruses).”

1.24 I. 134: a great sentence in which to use “microbiome science” rather than “microbial science”. Can you give MORE than just the single application of metagenomic structure/substructure insights?

“Applying Cartesian doubt to microbiome science has many applications to rethinking the rules of life for microbiomes, ranging from our view of metagenomic DNA substructure to the microbial species concept to the tools used to work with metagenomes.”

1.25 I. 138: Are you really arguing for a rethink of TAXONOMY? Or are you suggesting a radical revision of what taxonomy means (which is what I think you are arguing for), in which case I encourage you to articulate this more clearly. Do you mean a taxonomy of the community as an aggregate unit? What ideas can you touch on in the literature to illustrate this more clearly?

We have attempted to reduce the focus throughout the manuscript on taxonomy throughout the manuscript as well as here, as our goal extends beyond even just how we classify microorganisms. We additionally provide a few examples of algorithms that could be starting points for future development, as mentioned in point 1.14.

1.26 I. 140: There is a LOT of research in synthetic communities that I think builds to what you are talking about here, including both wet lab/experimental work and whole genome-based metabolic modeling (which can/should be reenvisioned from a metagenomic place). Are you suggesting something specific beyond the extensive work in this area at present? Clarity would help.

We have expanded this section to make it not only above more tools, but also how the interplay between theory and practice:

“Second, theory and empirical data collection (i.e. experimental practice) need to inform each other. Currently, the field’s assumptions constrain methodological development. If microbiological theory were less historically biased, further(33–35) wet lab techniques for operating on different units of microbial life could be developed, perhaps extending on current synthetic community work but relying more on enrichments of functional consortia or independent HGT elements (Figure 2C). Data interpretation methods are also needed, such as theoretical modeling(20) and algorithms operating on k-mers and microbiome metabolism.”

1.27 I. 143: Rather than a avoiding ‘universal gold standards’, perhaps what is more critical is defining the standards AND defining the underlying assumptions and justification for the standards in relation to core questions. Standards can facilitate progress, but can also impede progress. The sweet spot is figuring out what they are accomplishing one versus the other---which generally depends upon the question. Notable, your line 146-147 falls back into the box of genomic analyses, but as far as I understand the paper a central point is that microbiomes can and should be explored using non-genome-constrained analyses for our field to both break out of a constraining paradigm and make big advances in fundamental understanding.

We agree that gold standards should be generated in light of a specific question. We now further emphasize this point:

“Finally, “gold standards” must only be defined in the context of a particular research question, avoiding claims regarding universality, as doing so obscures assumptions that may be invalid in context (e.g. >95% sequence identity when comparing genes, “complete” genomes).”

1.28 I. 149: what is meant by the idea of `application' here? It doesn't seem to be a practical application that you are referring to, but rather the ideas/questions that are being addressed? And notable, you are not advocating a paradigm-free approach, but really you are suggesting an explicit alternative paradigm to provide a distinct and important new perspective on the organization and structure of microbiomes. Certainly this new paradigm will both contribute wonderful new insights, but will also suffer from its own assumptions, but the complementarity of these paradigms is ultimately (in my view) the strength of our science. It seem to me that you are calling out the prospect of knowledge based upon: i) a single paradigm; and/or ii) the lack of consideration of the assumptions implicit or explicit in the paradigm that is being used to organize knowledge; and/or iii) a lack of recognition of the explicit limitation of a single paradigm for capturing all understanding/knowledge of a microbiome; and iv) a lack of recognition that new forms of data can empower new paradigms that may displace, or may complement, or may augment, or may empower existing paradigms. It seems to me that this isn't an either/or but really strong keeping our eyes open to multiple perspectives. We can accommodate, integrate, contrast, and query multiple data types using distinct paradigms to form a more perfect understanding (knowledge), e pluribus unum for science? Out of many (or at least multiple) paradigms, can we find the best knowledge? I would encourage you to take a BIGGER VIEW to your closing paragraph to inspire the community to adopt a bigger vision for microbiome science.

We have aimed to strengthen our final paragraph in light of these comments, specifically emphasizing that we are not trying to eliminate assumptions, rather minimize them and through that generate new paradigms, which of course themselves will be stepped in assumptions. We write:

“Minimizing assumptions will allow microbiome scientists to uncover biology while contextualizing our current paradigm. Indeed, Cartesian doubt's true power is accommodating many different perspectives, not necessarily unveiling some grand truth, but rather adjusting reference points through complementary scientific lenses. The historical model of microbiology has gotten us extremely far, and its value cannot be overlooked. However, while we all stand on the shoulders of giants, it is occasionally prudent to glance down at the ground.”

1.29 Use MICROBIOMES instead of micro-organisms or microbiology in places where possible throughout (recognizing that sometimes microbiology or microorganisms makes more sense).

We have gone through the manuscript and updated accordingly.

1.30 Also, what about fungi? Are the arguments you make here equally valid for fungi within microbiomes? Can you state this clearly somewhere in the manuscript? We would prefer that all of our pieces be as inclusive as possible.

We now explicitly mention cross-kingdom microbes twice:

“Finally, considering beyond base-pair-order-focused approaches will facilitate alternative sequencing (e.g. Hi-C) into our understanding of metagenomic communities, their meta-phenotypes (e.g. colonization resistance), and their diversity (e.g. bacteria, fungi, and viruses).”

“No microbial genome (of any kingdom of life)...”

Reviewer comments:

Reviewer #1 (Comments for the Author):

2.1 This article provides a useful reminder for microbiome researchers to think outside of the "individual-organism" box, that is, to transition from thinking of a microbiome as the additive sum of individuals to thinking of it as a synergistic collective. The authors focus specifically on nucleic sequence data, as these widely available data are more prone to linkages to the originating cells than non-sequence-based data such as metabolomes. This article is harkening back to the original concept of metagenomes put forth by Jo Handelsman, which was to think of the community itself as a meta-organism with the metagenome representing the collective. This is a powerful concept and one that bears repeating, reminding and presenting from new viewpoints. This article is useful in that it highlights some of the hurdles that are distracting us from this goal. For example, the authors highlight the current emphasis on assembly, which is highly problematic in complex communities; these efforts are directing time and resources away from efforts to extract information from the collective and often result in incomplete and likely highly error-prone assemblies. The authors also highlight how the desire for standards, which are a mechanism for increasing data credibility and quality, can also enforce assumptions that limit the potential for new discoveries as to the way that DNA is moved, shared and changes within a microbiome.

Overall, the article is a useful thought-piece encouraging scientists to question all assumptions regarding how microbes live and function within communities and to incorporate this questioning in their approaches to analyzing metagenome data. Such critical thinking is fundamental to good science, but given the potential for new discoveries in microbiome science and our tendency to fall back on what we already know of microbes, the authors' advice to use Cartesian doubt is a useful reminder to researchers. This is particularly needed for microbiomes since most of our current understanding of microbes comes from observing pure cultures (not communities) in liquid culture (not biofilms) under ideal conditions for growth (not resource limited) and in a constant environment (not with environmental shifts characteristic of most natural habitats).

Hence, most of what we know may simply not be applicable in the microbiomes that we are studying.

We thank the Reviewer for their comments and were thrilled that they found merit in our work. We agree that our piece echoes Jo Handelsman, and we now cite this paper in the following passage:

“Gaps between metagenomics(Handelsman Jo 2004) and historical microbiology evidence why microbiome scientists should reconsider our core assumptions (Figure 1A)”

2.1. The authors assume that microbial genomes are circular. This assumption has been long dispelled, since some microbes have linear chromosomes (e.g., *Borrelia*, *Streptomyces*, *Rhodococcus*) and even both circular and linear chromosomes (e.g., *Agrobacterium*). I would recommend deleting the references to circular genomes on lines 64, 105 and 157, and modifying "circularized" to "complete" on line 108.

This was an oversight that we have now remedied, removing all references to circular genomes.

2.2 I agree that minimizing discarded data is critical (I find that this is true even with comparative genomics of closely-related strains, so it would definitely be true for a community). However, it is difficult to see how "pre-partitioning [reads] by length" would help minimize discarded data (line 81). This should be clarified.

We have removed the comments regarding pre-partitioning reads by length, as we agree with the Reviewer that it was confusing.

2.3 I was surprised that the authors did not discuss the approach of comparing metagenomic reads, in the absence of assembly, to databases. Lines 95-101 discuss finding patterns in the sequences, including conserved motifs, but deriving meaningful information from these patterns requires knowledge. A read-based rather than assembly-based approach is being taken by some researchers and somewhat avoids this pigeonholing of all reads into "genes". Are there hurdles to this approach that should be addressed in their recommendations on lines 134-147? For example, are there changes/expansions to our databases that would improve the quality or speed of doing read-based comparisons?

We agree that these are points worth broaching. Alignment to databases is a valid way to avoid forcing reads into a gene-based framework. We are particularly interested in the content of these databases themselves. For example, instead of marker genes or other genomic features that may be currently in them, what if databases instead consisted of MASH signatures or some to-be-discovered sequencing-based data type? These new data types could supplement those in our current databases.

To get at this point, we have mentioned this (though of course there is much more that could be said) in the text at the lines recommended by the Reviewer. In our paragraph on analytical

strategies, we indicate that databases (especially those indexed by unique data types) would be useful potentially:

“However, methods and data structures (e.g. a database indexing the Sequence-Read-Archive with colored de Bruijn graphs) should be selected carefully, as different questions mandate different tools.”

2.4 Lastly, I did not understand how wet lab approaches for microbial communities, which I presume means amplifying and preserving these communities for experimentally repeatable studies, could be an "extension" of precision editing technologies. Rather, I can envision that the precision editing technologies themselves could be an outcome of success with the wet lab approaches.

We agree that our description was unclear. Specifically, we aimed with this point to emphasize that theory (e.g. culture-based vs. sequencing-based microbiology) and empirical data collection should be iterative, such that one informs the other. Doing so, we hypothesize, could potentially yield wet lab tools that operate on different units of microbial life than we currently can perceive (e.g. communities or different sequences). We now clarify this in the text, removing the specific comment on precision editing, which we agree was distracting from the main point:

“Second, theory and empirical data collection (i.e. experimental practice) need to inform each other. Currently, the field’s assumptions constrain methodological development. If microbiological theory were less historically biased, further(33–35) wet lab techniques for operating on different units of microbial life could be developed, perhaps extending on current synthetic community work but relying more on enrichments of functional consortia or independent HGT elements (Figure 2C). Data interpretation methods are also needed, such as theoretical modeling(20) and algorithms operating on k-mers and microbiome metabolism.”

And a few minor editorial suggestions:

2.5 Line 24. Delete "or" - "...that neither search for [or] nor rely on..."

We have done so.

2.6 Line 41. The sentence needs a verb. Tensions between metagenomics and historical microbiology evidence [illustrate? Indicate?] why we should reconsider our core assumptions.

The sentence now reads:

“Gaps between metagenomics(4) and historical microbiology evidence why microbiome scientists should reconsider our core assumptions (Figure 1A)”

2.7 Line 65. Suggested revision "...not assuming the veracity of taxonomy, of genomes, or frankly of what we think we know about microbial life"

This sentence has been moved and reworked, it now reads:

“We propose similarly rethinking microbial communities via DNA sequencing, reimagining what microbial life may be instead of assuming what it is on the basis of existing taxonomy, microbial genomes, or other culture-centric paradigms.”

2.9 Line 81. Suggested revision "it may be worth pre-partitioning by length or another feature such as biological source" or "it may be worth pre-partitioning by features such as biological source".

We have removed this sentence.

Reviewer #2 (Comments for the Author):

3.1 The manuscript by Tierney and colleagues is an interesting Perspective article that encourages microbiologists to think differently about the field - to approach it from a DNA-centric approach rather than an organismal approach. The manuscript is couched in a philosophical framework calling researchers to use Cartesian doubt/radical doubt/radical skepticism. Overall, although I appreciate that this is a shorter format of manuscript, I felt the arguments were too abbreviated to be clear and did not adequately represent the current state of the field.

We thank the Reviewer for their comments and we hope that our revisions -- in which we cut out or revised much of the text that seemed to be confusing or over-abbreviated -- have clarified our points.

3.2 The title and manuscript seem to use "Cartesian doubt", "radical doubt", and "radical skepticism" interchangeably without a good solid definition the first time the phrases are used. I gather that "Cartesian doubt" is the most widely used term in philosophy. I would encourage the authors to use that phrase throughout and to give a simple definition the first time it is used. I needed to do a google search to understand the title.

We agree our vernacular was unclear, and we now define Cartesian doubt in the first sentence and no longer use other phrases interchangeably with it. We write:

“Cartesian doubt -- beginning with radical skepticism and moving forward with as few external assumptions as possible -- can be used to reconstruct our current practice of microbiome science, addressing biases and conflicts stemming from centuries of culture-based microbiology.”

3.3 The authors assume a very specialized audience throughout the manuscript that is highly attuned to techniques and issues that are commonly used with genomic and metagenomic sequencing. I consider myself someone that knows much of what the authors are talking about and still found much of the manuscript was way too much "insider baseball" (e.g. de bruijn graphs, Hi-C, exploding MAG problem, etc). The manuscript would really benefit from the

authors expanding their examples and being more clear about what they are arguing. The manuscript needs a thorough edit with the eyes of someone that is not so immersed in the field.

We want our manuscript to be relatable to as large an audience as possible, and so we are very grateful for this comment. We went through the text in detail, attempting to define phrases (like Cartesian doubt) where necessary and walk back overly technical language (eg de bruijn graphs) wherever possible. We removed most references to MAGs (including the exploding MAG problem). Due to the focus on sequencing technology, we did leave some of the terms in as we felt them necessary and reasonably broad (e.g. Hi-C), though we were sure now to emphasize that Hi-C is an example of sequencing genome spatial structure.

3.4 I was a bit lost throughout this manuscript as to what the authors were really saying. So many of the examples of problems in this Perspective are because of methods - missing the Archaea, junk DNA, etc. To the authors' first example of traditional microbiology, even Van Leeuwenhoek's predecessors missed the Bacteria because of the limits of technology. I would even argue that a microscope is not that far removed from software that processes a FASTA file. Van Leeuwenhoek had to grind lenses to magnify the samples using a very sensitive process that was not reproducible to other scientists at the time - many thought he was fabricating results! (doi: 10.1098/rstb.2014.0344) The parallels are similar with bioinformatics software for processing DNA sequences. To me, the problems are with picking the best methods and questioning the assumptions of those methods rather than with the underlying biological hypothesis we are applying.

We agree that our original draft could have used greater clarity in terms of our overall point. We hope the new one comes closer to communicating what we're saying. While we have revised the manuscript heavily, we additionally describe here our response to this comment:

First, to the Reviewer's point comparing a microscope to the software that processes a FASTA file: we see what they are saying, however we would argue that the fundamental tension highlighted by our piece is that at present, metagenomics attempts to impose a microscope's way of looking at things (or, more specifically, the view of life engendered by a culture-based microbiological framework) on to DNA data. In other words, we develop methods and software using "view" of microbiology that -- relative to DNA sequencing -- may not fully capture what is there.

Second, we agree with the Reviewer that it is critical to pick the best methods for a particular question (as we get at with the need to contextualize gold standards and in our additional clarifying statement on analyses and data structures), and we also agree that the assumptions of those methods should be questions. However, this also means you also need to question the underlying biological hypotheses stemming from those methods. For example, consider software built to process a FASTA file that assumes all reads can be mapped to reference genomes. Recall that the idea of a reference genome stems from culture-based microbiology. This software will likely throw out reads that do not map to anything (thereby missing potentially

useful data), and any conclusions drawn from it will be based in the paradigm from which it was built: again, that of culture-based microbiology.

Cartesian doubt, or, temporarily throwing away the assumptions that come along with grinding lenses and growing microbes, can potentially allow us to build software that looks at microbial communities in a different way.

3.5 Isn't the problem the authors are outlining with our use of methods? I'm not convinced that Science is not sufficiently self correcting. I think it sees through these problems. Unfortunately, the authors see Science as "half-empty" in that regard. As the authors rightly acknowledge there are plenty of examples of where our algorithms have been founded on incorrect assumptions. Yet the paragraph detailing this (L77-85) has few citations and the citations that are included describe methods. They might consider doi: 10.1126/science.1142490 as an example for where "contaminating bacterial DNA" was actually a Wolbachia genome embedded in an insect genome. Although they mention that we didn't know about Archaea until relatively late, we did find out about them through both cellular and molecular approaches. Even later we realized that we were missing phyla of Archaea through microscopy and molecular approaches (DOI: 10.1073/pnas.0914470107) - but Science self corrected itself in all of these examples. Again, I'm left wondering what the authors were really arguing for in this manuscript.

We thank the Reviewer for this comment and have provided some clarification below (in addition to overall reworking the manuscript to clarify our point).

First, we are not writing off the titanic effort biologists did exert to apply dry and wet lab approaches to, for example, discover archaea (or more recently, BORGs), or, as the Reviewer points out in a spectacular example, the Wolbachia/insect contamination case study, which we have now included in the manuscript. All we intend to say is that there may be even more undiscovered microbial biology that alternative paradigms for considering life -- like viewing its higher-order structure on the community level -- may unveil.

Second, we agree that science is self-correcting. Indeed, we would claim that questioning the hypotheses arising from historical methods (as we attempt to do here) is actually part of that course-correction process. As we state in the previous point (3.5), we are not raising issues with specific methods per se, rather we are indicating that deployment of methods built under a certain paradigm -- bioinformatic and otherwise -- may miss certain aspects of biology, or at least construe them in a biased manner.

3.6 I am generally confused by their claimed results of a "thought experiment". We know that linear bacterial (and viral) genomes exist (doi: 10.1111/j.1462-2920.2007.01328.x). I'm not sure that we could figure out codons from only DNA sequences without amino acid sequences. At a minimum, a citation should be provided here. Geneticists are already reconsidering what a gene is, without regard for metagenomics or even bacteria (doi: 10.1101/gr.6339607)

First, we agree it was a substantial oversight to describe complete genomes as circular, and we have since remedied that throughout the manuscript.

Regarding the Reviewer's point on recovering codons: while we suppose the only way to truly figure it out is do an experiment (could algorithm be trained to identify skews in the 64 possible 3-base-pair patterns in DNA), we have now included a recent preprint that indicate this, to us, that this likely is the case: <https://www.biorxiv.org/content/10.1101/2021.06.18.448887v1>.

We believe the Reviewer's example of geneticists reconsidering what a gene is as yet another excellent one, and we have included it in the manuscript. We additionally include another reference to this point now (<https://link.springer.com/article/10.1007%2Fs13752-012-0014-x>).

The paragraph in referenced by the Reviewer in this comment (3.6) aims, as they surmise, to simply indicate that Cartesian doubt applied specifically to the microbiome may not only raise -- but answer -- similar questions to those raised in this manuscript. In other words, there may be patterns in metagenomic sequences that could be identified that we haven't seen in addition to those that we may be able to recover (like codons).

3.7 I think the authors are also somewhat selective in their rejection of the cellular model of science. They discuss "tensions" (L41) but only indicate problems with the traditional approach. If there is a tension then there must be problems on both sides. The manuscript is very one-sided in its critique. Are there no benefits from the traditional approach? The paragraph starting at L119 seems to argue for treating communities as bags of genes, disregarding the importance of cellular envelopes. Although their argument for lack of importance of cellular structure is based on metabolism being widespread across a community, cellular structures could still be important. I also think of cellular structures being important for other characteristics like pathogenesis and toxin production. Do the authors see no benefit to considering cells as the atomic microbial level?

We thank the Reviewer for this comment in particular. As a group of environmental and human microbiologists who use (and cherish) traditional isolation and culturing as well as computational approaches, we naturally see tremendous value to the cellular model of life. Where would we be without it? Indeed, as the Reviewer points out in another comment (3.4), we only know the issues with our current paradigm because we have in many instances been able to overcome them. Our goal is to raise the question of the unknown -- to what degree is the existing paradigm keeping us "searching under the lamppost," so to speak? Given that we have found exceptions to the rule (once again considering the example of contamination in an insect genome turning out to be *Wolbachia*), we hypothesize that a new paradigm reliant on sequencing data alone may uncover even more exciting new biology, while, of course, also coming with its own limitations and framing.

We now explicitly emphasize the value of the cellular model in the text in the second-to-last sentence of the text:

“The historical model of microbiology has gotten us extremely far, and its value cannot be overlooked.”

3.8 The final paragraph starts with, "Allowing our view of microbial communities to be guided by application and not paradigm-bias..." Yet, I would argue that Science moves forwards because we have paradigms - hypotheses - that are continually refined. If everything is operationally or practically defined, Science cannot move forward or self correct.

We agree -- science moves forward because of paradigms, however its progress can be shifted (if not halted in some ways) if these paradigms become too rigid. This is the exact point we want to make with our (now revised) description of the danger of gold standards, and the need for them to be contextualized. In addition to removing the quoted phrase above from the manuscript, the second to last paragraph (where we discuss gold standards) is now clearer, and we write:

“Finally, “gold standards” must only be defined in the context of a particular research question, avoiding claims regarding universality, as doing so obscures assumptions that may be invalid in context (e.g. >95% sequence identity when comparing genes, “complete” genomes).”

September 3, 2021

Dr. Chirag J Patel
Harvard Medical School
Biomedical Informatics
10 Shattuck St
Boston, MA 02115

Re: mSystems00574-21R1 (Using Cartesian doubt to build a sequencing-based view of microbiology)

Dear Dr. Chirag J Patel:

Thank you for submitting your manuscript to mSystems. We have completed our review and I am pleased to inform you that, in principle, we expect to accept it for publication in mSystems. However, acceptance will not be final until you have adequately addressed the reviewer comments.

Braden and Co-authors:

Great improvements to the manuscript! Really an excellent job. I also greatly appreciate your detailed and thoughtful response to reviewer's comments. In addition to some very minor text edits (made directly onto the scanned version that is attached), I have a few other suggestions below. Pending your review and incorporation (or rational rejection) of these suggestions, I see moving this quickly to PUBLISH!

1. L. 23: can a statement be incorrigible?

2. L. 28-30: I wonder if the final clause could be less win/lose. Perhaps something like: "A view of microbial communities based on sequence data unconstrained by current notions is likely to provide novel insights into metagenomic structure that may more closely represent (capture?) functional biology within the microbiome."

3. Throughout, check the format of the references/literature cited within the body of the text to correspond to mSystems format.

4. The second paragraph of the piece is the very most important one in laying out your points. I offer the following suggestions to take one more stab at increasing the precision of your great arguments and ideas!

5. L. 54: I would recommend inserting part of one of your later sentences in the middle of this line. Specifically, I would recommend inserting: "Overall, microbiologists have constructed paradigms to cohere with data generated from pure culture studies, but have not constructed novel (novel?) paradigms for microbiomes based upon 'omics-based sequence data and unconstrained by pure culture paradigms."

6. L. 57: Maybe substitute "This can constrain our understanding of the biology of complex microbiomes." for "These constrain our understanding of microbiology"

7. L. 59: I'm not sure what 'metagenome-spanning signals' refer to? HGT and evolutionary drift are important points here, but I am not sure that the casual reader will immediately understand 'metagenome-spanning signals'. Are HGT and evolutionary drift actually more explicit genome (individual genome)-spanning signals rather than metagenome-spanning signals? I think this sentence makes a very important point---that the traditional assembly and binning lacks a clear or integrated recognition of the signature/impacts of horizontal gene transfer and evolutionary drift within the microbiome.

8. L. 63-65: Do you mean that the functional roles of genes with similar sequences are often defined through global percent identity cutoffs, despite sequence not necessarily or consistently correlating to function?

9. L. 65-66: Finally, bio- and geochemical reactions exist in multiple spatial and temporal structures that may or may not be membrane bound within discrete cells.

10. L. 66-end of paragraph: Overall, a metagenome-centered approach based upon sequence data and unconstrained by pure-culture-based paradigms provides an exciting opportunity to rethink assumptions about the (functional?) organization of microbial life, and promises to transform our understanding of microbiomes.

11. L. 73/74: recommend changing to "...versus the paradigms used to interpret those data".

12. L. 91: add clause to end of sentence: "...network strategies to identify significant patterns in microbiome sequence data (19-24)."

13. L. 91/92: recommend changing to "These approaches will support (advance?) analyses unconstrained...."

14. L. 94-95: recommend changing to "...different questions may mandate different tools and approaches."

15. L. 98: do you need quotation marks around the word sequence?

16. L. 108-109: Could you strengthen the opening sentence of this paragraph, and perhaps make it simpler? For example "Analyses of microbiome data based upon existing analytical paradigms may (are?) also be limited in their capacities to capture temporal variation within genomes".

17. L. 109 parenthetical clause: (or genome within any kingdom of life)

18. L. 111: rather than WOULD yield, how about WILL yield?

19. L. 114: WILL align rather than WOULD align?

20. L. 136: "...will facilitate incorporation of alternative sequencing data (e.g. Hi-C)...."

21. L. 159: "contextualizing our current paradigm" has unclear meaning here. ".....will add complementary insight to current paradigm(s?), adding depth/richness to our understanding of the (the functional? Ecological? Organizational? Evolutionary?

Other?) organization of complex microbiomes. "

22. I read and re-read the final sentence multiple times. While I like the broad analogy, I struggled with whether or not the sentence as written really nailed the idea. Is it prudent to glance at the ground, or is it prudent to consider the ground upon which the giants are standing? And then I wondered if I were totally over-thinking it. But you might wordsmith the final sentence or shop it among some of your labmates to see if they `get' the idea that you are seeking to express in the closing. FINALLY, I appreciate the reworking of the figures. However, I think that you have the potential to create an even more high-impact image by merging the two figures into one rich `infographic' that captures your key ideas in a more explicit flow chart. This could be a useful, single high-impact image that weaves together your entire thesis. This would build upon your existing elements (1A, 1B, etc.), but add a few simple elements to make the whole. HOWEVER, I WANT TO BE TOTALLY CLEAR IN LEAVING THIS TO YOUR DISCRETION! MOREOVER, I WANT TO SHARE THE COMMENT OF ONE OF THE OTHER CO-EDITORS I ASKED FOR A READ-THROUGH OF YOUR PAPER, AND HE REALLY LIKED THE EXISTING FIGURES (SEE COMPLETE COMMENTS AT THE END OF THIS DOCUMENT).

The starting point/foundation is image 1B: the microbiome a complex collection of organisms with multiple mechanisms for generating sequence diversity and heterogeneous populations coexisting in space and time. Consider inserting a new box (3A), which is the game-changer: SEQUENCING (can also be valuable place where you can note this is RNA, DNA, genome, transcriptome, whatever-a key point wrt versatility). Then the figure highlights the 2 parallel pathways to discovery/understanding: 1A represents the historical (constrained) approach; new box 3B defines the new, unconstrained approach, a fundamentally different pathway. 1A (constrained by what we know) leads to 1C as the analytical pathway. 3B suggests 2A: wide open, pattern recognition. Then, the game changer, an unconstrained model leads to 2A as an analytical pathway. The outcomes of these two pathways are a new box (3C, how shifts in relative abundance of taxa/bins, modest information on genomic variation within a bin), or 2B, and amazing, mind-blowing overview of the metagenome distinct from bins and taxa. 2C can still exist within the intermediate space, and could emphasize the differences in analytical outcomes and especially the potential to use experimental approaches in combination with unconstrained metagenomic analyses to uncover fundamental information on the organization and variation in the metagenome as a function of consortium/condition/etc. that would be missed using the historical lens. I am wondering if whether some type of merging of the two figures that allows a direct contrast could be much more high-impact and likely to be useful in broader contexts. The lens DOES determine the analytical approaches, which determine the outcomes in models and understanding. That is a key part of your thesis. And, of course, that both 3C and 2B are valuable and offer complementary insights.

BUT THIS IS ALL YOUR CALL, I was just seeking a grand synthesis. 😊

Finally, as mentioned, I had one of our other coeditors provide me with his `big picture' thoughts on your contribution. Here are his comments, in full (see below). I hope these make you feel great, and motivate you to get these last final tweaks wrapped up and then we will have it PUBLISHED!!

"My broad overview impression of this Perspective is that it will be a catalyst for many excellent conversations and debates. It is exciting and provocative in productive/thoughtful ways. I think the reviewer comments were very helpful and they addressed them adequately. The problem is clearly laid out in the first few paragraphs, and they provide some helpful examples along the way to illustrate their perspective. I also really like the figures. They are clear, rich, and help anchor the text.

One issue is that they suggest many mandates for the field, but don't fully explain them. For example, the entire paragraph starting in Line 140 is a list of "to-dos" for the field, but they don't fully explain how to accomplish each of those tasks. I bet that is because there just isn't enough space to do that in these short pieces. I think that is entirely OK for a short Perspective piece.

I think it is nearly ready for publication and will be a standout piece in our collection."

Thanks to you and your team for all your work on this piece. I agree with the co-editor that this will be a wonderful piece in our collection.

PLEASE let me know if you have any questions or comments, our goal now is to get this one DONE! I am assuming that you will find most of these edits straightforward, and will be happy to move to full acceptance upon your consideration of the suggestions.

Best wishes, have a great weekend!

Linda

Preparing Revision Guidelines

To submit your modified manuscript, log onto the eJP submission site at <https://msystems.msubmit.net/cgi-bin/main.plex>. Go to

Author Tasks and click the appropriate manuscript title to begin the revision process. The information that you entered when you first submitted the paper will be displayed. Please update the information as necessary. Here are a few examples of required updates that authors must address:

Sincerely,

Linda Kinkel

Editor, mSystems

Journals Department
Reviewer comments:

Dear Dr. Kinkel and the Editorial Team,

We are truly thrilled that you found our manuscript improved, and once again we are grateful for the detailed comments. As before, in addition to a clean, updated copy of the text, we are including a tracked changes version of the manuscript and a response to your comments in this document. We addressed each of the comments as best as possible, and we had to modify small other portions of the text (a word here or there) to ensure that we were under the word limit.

Thank you again for giving us the chance to participate in your collection and publish in mSystems. This has truly been a delightful experience and we are so excited to read the other pieces.

Sincerely, and on behalf of the authors,

Braden T Tierney and Chirag J Patel

Braden and Co-authors:

Great improvements to the manuscript! Really an excellent job. I also greatly appreciate your detailed and thoughtful response to reviewer's comments. In addition to some very minor text edits (made directly onto the scanned version that is attached), I have a few other suggestions below. Pending your review and incorporation (or rational rejection) of these suggestions, I see moving this quickly to PUBLISH!

1. L. 23: can a statement be incorrigible?

We understand incorrigibility to be a philosophical term referring to statements that are true by virtue of them being believed (e.g. "I think therefore I am", <https://plato.stanford.edu/entries/certainty/>). While we do want to make this piece as accessible as possible and accept this is a perhaps niche definition, we feel that it is particularly relevant to motivating/understanding Cartesian Doubt and therefore would prefer, if possible, to leave it in.

2. L. 28-30: I wonder if the final clause could be less win/lose. Perhaps something like: "A view of microbial communities based on sequence data unconstrained by current notions is likely to provide novel insights into metagenomic structure that may more closely represent (capture?) functional biology within the microbiome."

We agree this could be less win/lose, especially regarding the "more closely represents functional biology" statement. We've modified accordingly:

We propose that a view of microbial communities based in sequencing will engender novel insights into metagenomic structure and may capture functional biology not reflected within the current paradigm.

3. Throughout, check the format of the references/literature cited within the body of the text to correspond to mSystems format.

We have done so.

4. The second paragraph of the piece is the very most important one in laying out your points. I offer the following suggestions to take one more stab at increasing the precision of your great arguments and ideas!

5. L. 54: I would recommend inserting part of one of your later sentences in the middle of this line. Specifically, I would recommend inserting: "Overall, microbiologists have constructed paradigms to cohere with data generated from pure culture studies, but have not constructed novel (novel?) paradigms for microbiomes based upon 'omics-based sequence data and unconstrained by pure culture paradigms."

See point 10.

6. L. 57: Maybe substitute "This can constrain our understanding of the biology of complex microbiomes." for "These constrain our understanding of microbiology"

This sentence now reads:

“This can constrain our understanding of microbiome biology”

We modified from the recommendation to keep the text within the 1500 word limit.

7. L. 59: I'm not sure what 'metagenome-spanning signals' refer to? HGT and evolutionary drift are important points here, but I am not sure that the casual reader will immediately understand 'metagenome-spanning signals'. Are HGT and evolutionary drift actually more explicit genome (individual genome)-spanning signals rather than metagenome-spanning signals? I think this sentence makes a very important point--- that the traditional assembly and binning lacks a clear or integrated recognition of the signature/impacts of horizontal gene transfer and evolutionary drift within the microbiome.

We agree metagenome-spanning signals not just jargon, but also perhaps jargon we were making up on the spot. We considered replacing with “genome-spanning,” but were concerned this may connote “a signal that acts across a single organism’s genome.” We instead now write “ecosystem-spanning,” in the hope that this will be clearer.

8. L. 63-65: Do you mean that *the functional roles of genes with similar sequences* are often defined through global percent identity cutoffs, despite sequence not necessarily or consistently correlating to function?

We agree the word “identical” was not clear enough here and have updated as recommended (we were thinking “identical” as in “non-redundant gene catalog” construction initially).

9. L. 65-66: Finally, bio- and geochemical reactions exist in multiple spatial and temporal structures that may or may not be membrane bound within discrete cells.

We have updated accordingly.

10. L. 66-end of paragraph: Overall, a metagenome-centered approach based upon sequence data and unconstrained by pure-culture-based paradigms provides an exciting opportunity to rethink assumptions about the (functional?) organization of microbial life, and promises to transform our understanding of microbiomes.

We were torn between this point and point 5, as we felt as if we moved the current sentence *and* changed it as recommended, it would read more like a concluding sentence and be out of place. As a result, we decided to choose between keeping it at the end of the paragraph and changing it or moving to earlier in the paragraph. We chose to do the former while also slightly modifying the beginning. We change the earlier portion of the paragraph by writing:

“why microbiome scientists should reconsider our core assumptions (**Figure 1A**) -- though the field should not adopt an ahistorical view. Rather, researchers should acknowledge that contemporary analyses can be biased by prior, culture-based experiments.”

We then left the concluding sentence where it was but modified it to capture the sentiment of the recommended edits.

“Overall, microbiologists constructed paradigms to cohere with pure cultures; a sequencing-centered approach to metagenomics unconstrained by pure-culture-based paradigms provides an exciting opportunity to rethink assumptions about the organization of microbial life.”

11. L. 73/74: recommend changing to “...versus the paradigms used to interpret those data”.

We have made a slightly modified version of this change (referring to microbial communities instead of data):

“Cartesian doubt can address epistemological conflicts between observations (i.e. raw data) from microbial communities and the paradigms (i.e. theory) used to interpret them”

12. L. 91: add clause to end of sentence: “...network strategies to identify significant patterns in microbiome sequence data (19-24).”

We opted not to make this change, as we felt that while the additional detail would be necessary, the cost of these particular added words (being right at the word limit) outweighed their need. We of course can change this if needed.

13. L. 91/92: recommend changing to “These approaches will support (advance?) analyses unconstrained.....”

We have made this change.

14. L. 94-95: recommend changing to ““...different questions may mandate different tools and approaches.”.

We wrote:

“different questions mandate different approaches,” shortening the recommendation to fit under the word limit.

15. L. 98: do you need quotation marks around the word sequence?

We do not, and have removed them.

16. L. 108-109: Could you strengthen the opening sentence of this paragraph, and perhaps make it simpler? For example “Analyses of microbiome data based upon existing analytical paradigms may (are?) also be limited in their capacities to capture temporal variation within genomes”.

We write (once again looking to shave off extra words) “Analyses based upon existing paradigms may also be limited in their capacities to capture genomic temporal variation”

17. L. 109 parenthetical clause: (or genome within any kingdom of life)

We have made this change.

18. L. 111: rather than WOULD yield, how about WILL yield?

We have made this change.

19. L. 114: WILL align rather than WOULD align?

We have made this change.

20. L. 136: "...will facilitate incorporation of alternative sequencing data (e.g. Hi-C)..."

We have made this change.

21. L. 159: "contextualizing our current paradigm" has unclear meaning here. ".....will add complementary insight to current paradigm(s?), adding depth/richness to our understanding of the (the functional? Ecological? Organizational? Evolutionary? Other?) organization of complex microbiomes. "

We modified this sentence to the point where it now reads:

"Minimizing assumptions will add complementary insight to the current paradigm while adding richness to our understanding of the functional organization of complex microbiomes."

22. I read and re-read the final sentence multiple times. While I like the broad analogy, I struggled with whether or not the sentence as written really nailed the idea. Is it prudent to glance at the ground, or is it prudent to consider the ground upon which the giants are standing? And then I wondered if I were totally over-thinking it. But you might wordsmith the final sentence or shop it among some of your labmates to see if they `get` the idea that you are seeking to express in the closing.

This was tough -- this sentence is one that I (Braden) am fairly attached to, however, we (the authors) do agree it could be clearer and have run it by a couple of outside parties. We have modified it ever so slightly:

"However, while we all stand on the shoulders of giants, it is occasionally prudent to consider the ground beneath our feet."

23. FINALLY, I appreciate the reworking of the figures. However, I think that you have the potential to create an even more high-impact image by merging the two figures into one rich `infographic` that captures your key ideas in a more explicit flow chart. This could be a useful, single **high-impact** image that weaves together your entire thesis. This would build upon your existing elements (1A, 1B, etc.), but add a few simple elements to make the whole. **HOWEVER, I WANT TO BE TOTALLY CLEAR IN LEAVING THIS TO YOUR DISCRETION! MOREOVER, I WANT TO SHARE THE COMMENT OF ONE OF THE OTHER CO-EDITORS I ASKED FOR A READ-THROUGH OF YOUR PAPER, AND HE REALLY LIKED THE EXISTING FIGURES (SEE COMPLETE COMMENTS AT THE END OF THIS DOCUMENT).**

The starting point/foundation is image 1B: the microbiome a complex collection of organisms with multiple mechanisms for generating sequence diversity and heterogeneous populations coexisting in space and time. Consider inserting a new box (3A), which is the game-changer: SEQUENCING (can also be valuable place where you can note this is RNA, DNA, genome, transcriptome, whatever—a key point wrt

versatility). Then the figure highlights the 2 parallel pathways to discovery/understanding: 1A represents the historical (constrained) approach; new box 3B defines the new, unconstrained approach, a fundamentally different pathway. 1A (constrained by what we know) leads to 1C as the analytical pathway. 3B suggests 2A: wide open, pattern recognition. Then, the game changer, an unconstrained model leads to 2A as an analytical pathway. The outcomes of these two pathways are a new box (3C, no hum shifts in relative abundance of taxa/bins, modest information on genomic variation within a bin), or 2B, and amazing, mind-blowing overview of the metagenome distinct from bins and taxa. 2C can still exist within the intermediate space, and could emphasize the differences in analytical outcomes and especially the potential to use experimental approaches in combination with unconstrained metagenomic analyses to uncover fundamental information on the organization and variation in the metagenome as a function of consortium/condition/etc. that would be missed using the historical lens. I am wondering if whether some type of merging of the two figures that allows a direct contrast could be much more high-impact and likely to be useful in broader contexts. The lens DOES determine the analytical approaches, which determine the outcomes in models and understanding. That is a key part of your thesis. And, of course, that both 3C and 2B are valuable and offer complementary insights.

BUT THIS IS ALL YOUR CALL, I was just seeking a grand synthesis. □

Thank you so much for your thoughtful consideration of these new figures. We considered both your comments and your colleague's comments deeply as a team. The tradeoff appears to lie, at the moment, between a single clear figure that demonstrates the "old" vs the "new" in a 1 unified image vs the detail provided by splitting this comparison into two separate images. After thinking hard about it, we feel, overall, that we want to keep the level of detail for scientists to really dig their mental teeth. We feel the current figures enable this, especially given how (at least in our opinion) the first figure not only shows the modern paradigm, but how the "real world" data of metagenomics is compressed into it.

That said, we really do like the "grand unified thesis" idea. Would there be any other potential spot to place a more "flow chart" figure like the one you recommended? A graphical abstract perhaps? Or we could see it being quite useful in a more public-facing setting (e.g. a twitter, etc) to really, as you say, hammer home the key goal.

24. Finally, as mentioned, I had one of our other coeditors provide me with his 'big picture' thoughts on your contribution. Here are his comments, in full (see below). I hope these make you feel great, and motivate you to get these last final tweaks wrapped up and then we will have it PUBLISHED!!

"My broad overview impression of this Perspective is that it will be a catalyst for many excellent conversations and debates. It is exciting and provocative in productive/thoughtful ways. I think the reviewer comments were very helpful and they addressed them adequately. The problem is clearly laid out in the first few paragraphs, and they provide some helpful examples along the way to illustrate their perspective. I also really like the figures. They are clear, rich, and help anchor the text.

One issue is that they suggest many mandates for the field, but don't fully explain them. For example, the entire paragraph starting in Line 140 is a list of "to-dos" for the field, but they don't fully explain how to accomplish each of those tasks. I bet that is because there

just isn't enough space to do that in these short pieces. I think that is entirely OK for a short Perspective piece.

I think it is nearly ready for publication and will be a standout piece in our collection.”

As we have said before, we are delighted by both your and your colleague's enthusiasm for our ideas and writing. We agree with the point raised here that our mandates read more like a list of demands sans clear explanation. They were accurate in surmising that this was because of the tight word limit, which as you can see we are right up against. We are glad that this limitation is acceptable, however, and we hope that the motivation for these particular 3 points can at least partially be understood by the earlier paragraphs in the manuscript. We attempted to make some edits adjusting this section but were unable to do so while staying under the word limit and not weakening the writing in the paragraph. As a result, we hope that keeping this the same will be satisfactory going forward.

September 23, 2021

Dr. Chirag J Patel
Harvard Medical School
Biomedical Informatics
10 Shattuck St
Boston, MA 02115

Re: mSystems00574-21R2 (Using Cartesian doubt to build a sequencing-based view of microbiology)

Dear Dr. Chirag J Patel:

Great job on the revisions! This is a really fine addition to our collection, thank you to you and your full author team for all your thoughtful work.

I should've mentioned that, at this point, we now have some flexibility in the text length (one of our author teams had to bow out). Thus, if you feel strongly and wish to revisit any spots where you were very careful to keep under the text limit, and wish to add text, you can certainly feel free to do so. However, I completely understand if you are satisfied with the manuscript as it stands. This is an excellent contribution.

FINALLY, I do think that there could be room for a unified figure-if you are interested, I am happy to reach out to find out if this could fit within a graphical abstract or an image associated with the abstract, twitter certainly, but perhaps even as art/an image for a/the virtual issue over? Please let me know if you would like me to pursue any of these ideas--I am happy to do so, but this is your call!!

Your manuscript has been accepted, and I am forwarding it to the ASM Journals Department for publication. For your reference, ASM Journals' address is given below. Before it can be scheduled for publication, your manuscript will be checked by the mSystems senior production editor, Ellie Ghatineh, to make sure that all elements meet the technical requirements for publication. She will contact you if anything needs to be revised before copyediting and production can begin. Otherwise, you will be notified when your proofs are ready to be viewed.

As an open-access publication, mSystems receives no financial support from paid subscriptions and depends on authors' prompt payment of publication fees as soon as their articles are accepted. =

Publication Fees:

We recognize that the video files can become quite large, and so to avoid quality loss ASM suggests sending the video file via <https://www.wetransfer.com/>. When you have a final version of the video and the still ready to share, please send it to Ellie Ghatineh at eghatineh@asmusa.org.

Sincerely,

Linda Kinkel
Editor, mSystems

Journals Department
Phone: 1-202-942-9338